# QUASI-EQUIVARIANT METANETWORKS

**Viet-Hoang Tran**[*]
Department of Mathematics
National University of Singapore
hoang.tranviet@u.nus.edu

**An Nguyen**[*]
FPT Software AI Center
Vietnam
annt68@fpt.com

**Benoît Guérand**
Department of Mathematics
National University of Singapore
benoit.guerand@u.nus.edu

**Thieu N. Vo**
Department of Mathematics
National University of Singapore
thieuvo@nus.edu.sg

**Tan M. Nguyen**
Department of Mathematics
National University of Singapore
tanmn@nus.edu.sg

## ABSTRACT

Metanetworks are neural architectures designed to operate directly on pretrained weights to perform downstream tasks. However, the parameter space serves only as a proxy for the underlying function class, and the parameter-function mapping is inherently non-injective: distinct parameter configurations may yield identical input-output behaviors. As a result, metanetworks that rely solely on raw parameters risk overlooking the intrinsic symmetries of the architecture. Reasoning about functional identity is therefore essential for effective metanetwork design, motivating the development of equivariant metanetworks, which incorporate equivariance principles to respect architectural symmetries. Existing approaches, however, typically enforce strict equivariance, which imposes rigid constraints and often leads to sparse and less expressive models. To address this limitation, we introduce the novel concept of quasi-equivariance, which allows metanetworks to move beyond the rigidity of strict equivariance while still preserving functional identity. We lay down a principled basis for this framework and demonstrate its broad applicability across diverse neural architectures, including feedforward, convolutional, and transformer networks. Through empirical evaluation, we show that quasi-equivariant metanetworks achieve good trade-offs between symmetry preservation and representational expressivity. These findings advance the theoretical understanding of weight-space learning and provide a principled foundation for the design of more expressive and functionally robust metanetworks.

## 1 INTRODUCTION

Modern problem-solving increasingly relies on neural networks, which encode vast amounts of information within their trainable parameters during learning, ranging the application from computer vision (Huang et al., 2020; Krizhevsky et al., 2012; He et al., 2015), natural language processing (Vaswani et al., 2017; Rumelhart et al., 1986; Hochreiter & Schmidhuber, 1997; DeepSeek-AI et al., 2025), and nature science (Raissi et al., 2019; Jumper et al., 2021). While these parameters capture rich knowledge, accessing and interpreting it remains a challenge.

---

[*]Equal Contribution
    Please correspond to: hoang.tranviet@u.nus.edu and tanmn@nus.edu.sg

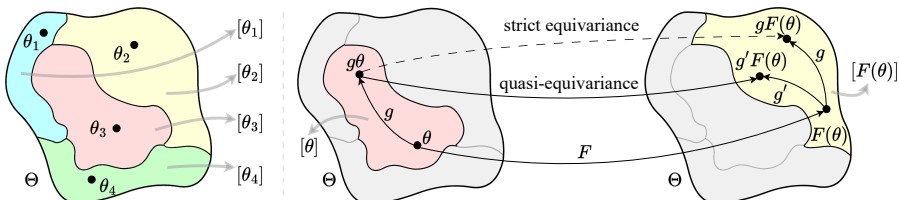

Figure 1: *(Left)* Illustration of the partition of parameter space into functional equivalence classes. *(Right)* Illustration of the quasi-equivariance property and its distinction from strict equivariance.

**Metanetworks.** Metanetworks were introduced to analyze and process other neural networks by treating their weights, gradients, and sparsity patterns as structured inputs. Early work focused on evaluating their generalization and revealing properties of neural network behavior (Baker et al., 2018; Eilertsen et al.; Unterthiner et al., 2020; Schürholt et al., 2021; 2022a;b). Common strategies include flattening parameters or extracting statistics before feeding them into multi-layer percep-trons (MLPs) (Unterthiner et al., 2020; Dupont et al., 2022; Luigi et al.). Beyond these foundations, metanetworks have been applied to extracting structure from implicit representations (Müller et al., 2023; Stanley, 2007; Mildenhall et al., 2021), developing learnable optimizers (Bengio et al., 2013; Runarsson & Jonsson, 2000; Andrychowicz et al., 2016; Metz et al., 2022), performing model edit-ing (Sinitsin et al., 2020; Cao et al., 2021; Mitchell et al., 2022), evaluating policies (Harb et al., 2020), and enabling Bayesian inference (Sokota et al., 2021). Nevertheless, designing metanetworks remains challenging due to the complexity and high dimensionality of the underlying structures.

**Functional Equivalence.** A major challenge in designing metanetworks lies on how to capture functional equivalence - the fact that multiple distinct parameter configurations can realize the same input-output function (Allen-Zhu et al., 2019; Belkin et al., 2019; Du et al., 2019; Frankle & Carbin, 2019; Novak et al., 2018). This problem was first posed by Hecht-Nielsen (Hecht-Nielsen, 1990). A key observation is that swapping two hidden units in an MLP leaves its input–output mapping unchanged, provided their outgoing connections are permuted accordingly (Allen-Zhu et al., 2019; Du et al., 2019; Frankle & Carbin, 2018; Belkin et al., 2019; Neyshabur et al., 2018). For the same class of MLPs, Fefferman & Markel (1993) established a stronger result: the input–output mapping of an MLP with tanh activations uniquely determines both its architecture and its weights, up to permutations and sign flips. Subsequent work extended these identifiability results to broader MLP settings (Albertini & Sontag, 1993b;a; Mai & Lampert, 2020; Chen et al., 1993; Kurkova & Kainen, 1994) and, in parallel, to convolutional neural networks (CNNs) (Brea et al., 2019; Novak et al., 2018; Mai & Lampert, 2020; Tran et al., 2024a; Vo et al., 2025).

**Equivariant Metanetworks.** Building on the insight from permutation invariance in neural net-works, researchers have developed permutation-equivariant metanetworks (Navon et al., 2023; Zhou et al., 2024a; Kofinas et al., 2024; Zhou et al., 2024b), which naturally account for neuron reorder-ing within hidden layers. More recent architectures extend beyond permutation equivariance by incorporating additional symmetries such as scaling and sign changes (Kalogeropoulos et al., 2024; Tran et al., 2024a; Vo et al., 2025). Furthermore, recent works (Tran et al., 2025b; Knyazev et al., 2025; Tran et al., 2025f) have characterized the maximal symmetry groups of Multihead Attention and Mixture-of-Experts, establishing necessary and sufficient conditions for functional equivalence, offering new insights into the structural properties of Transformer and Mixture-of-Experts weights.

Despite advances in metanetwork design, most approaches enforce strict equivariance at the level of individual weights. However, the true goal is not to preserve the weights themselves, but to capture the *functions* they implement–that is, to respect the *functional equivalence classes* defined by the parameters. In this context, relaxed or approximate equivariance has emerged in deep learning to handle imperfect symmetries in real-world data. Early approaches include weight-relaxed convolu-tions (Wang et al., 2022), soft constraints via multitask losses (Elhag et al., 2024; Pertigkiozoglou et al., 2024), G-biases for group convolutions (Wu et al., 2025), and extensions to $E(3)$-equivariant graph networks (Hofgard et al., 2024). On the theoretical side, Kaba & Ravanbakhsh (2023) and Huang et al. (2022) formalized relaxed equivariance and analyzed its bias–variance trade-offs. To-gether, these insights motivate a class of metanetwork architectures that relax strict weight-level equivariance, enabling more flexible representations of functional symmetries.

**Contributions.** Building on this motivation, we introduce a framework for *quasi-equivariant metanetworks*–a novel paradigm that relaxes strict equivariance to balance symmetry preservation with representational flexibility. The paper is organized as follows:

1. In Section 2, we examine the parameter space of a parameterized function, characterize its associated symmetry group, and introduce the formal notion of maximality within symmetry groups, establishing a direct connection to Functional Equivalence.

2. In Section 3, we examine the sufficiency of strict equivariance for metanetwork design. Building on this, we introduce quasi-equivariance, which enables metanetworks to overcome the limitations of strict equivariance while still maintaining functional identity.

3. In Section 4, we present a general framework for quasi-equivariant metanetworks and demonstrate its application to feedforward neural networks and multihead attention.

4. In Section 5, we integrate the framework into existing metanetworks. Experiments on multiple metanetwork benchmarks show that this layer enhances performance considerably while incurring only a slight increase in the number of parameters.

Supplementary materials, including a comprehensive notation table, theoretical derivations, detailed proofs, and experimental setups, are provided in the Appendix.

## 2 PRELIMINARIES ON EQUIVARIANT METANETWORKS

In this section, we present the details of the parameter space of a parameterized function, its associated symmetry group, and introduce the formal notion of maximality in symmetry groups, which connects directly to the concept of Functional Equivalence (FE).

### 2.1 PARAMETER SPACE OF A PARAMETERIZED FUNCTION AND ITS SYMMETRY GROUP

**Parameter space.** Let $f(\cdot; \theta)$ be a function parameterized by $\theta \in \Theta = \mathbb{R}^{\dim}$. The set $\Theta$ is called the *parameter space* (or *weight space*) of $f$. Assume a group $G$ acts on $\Theta$. For each $\theta \in \Theta$, we define the set of parameter vectors that yield functionally equivalent models:

$$[\theta] := \{\bar{\theta} \in \Theta \mid f(\cdot; \bar{\theta}) = f(\cdot; \theta)\} \subseteq \Theta. \tag{1}$$

The parameter space serves merely as a proxy for the function class, and the mapping $\theta \mapsto f(\cdot; \theta)$ is non-injective, as distinct parameter configurations can yield identical behaviors. This phenomenon is illustrated in Figure 1. FE thus focuses on characterizing the sets $[\theta]$. Explicitly enumerating all such sets is impractical. A more systematic approach is to view these equivalence classes as orbits under a group action on $\Theta$, naturally leading to the notion of the *symmetry group* of $f$.

**Symmetry group.** Consider a group $G$ acting on the space $\Theta$. For $\theta \in \Theta$, the $G$-orbit of $\theta$ is defined as $G\theta := \{g\theta \mid g \in G\} \subseteq \Theta$. We now introduce the following definition.

**Definition 2.1** (Symmetry Group). A group $G$ is called a *symmetry group* of the function $f$ if $G\theta \subseteq [\theta]$ for all $\theta \in \Theta$. Equivalently, for every $g \in G$ and $\theta \in \Theta$, one has $f(\cdot; g\theta) = f(\cdot; \theta)$.

The phrase "a symmetry group" acknowledges that multiple such groups may exist. In particular, every subgroup of a symmetry group is itself a symmetry group. Our goal is to represent the equivalence classes $[\theta]$ using $G$-orbits. To build intuition, we present two following observations.

*First observation.* Consider the function $f(\cdot; a, b) \colon \mathbb{R} \to \mathbb{R}$, defined by $x \mapsto abx$, parameterized by $\theta = (a, b) \in \mathbb{R}^2 = \Theta$. It is straightforward to see that $(a, b)$ and $(\bar{a}, \bar{b})$ yield the same function if and only if $ab = \bar{a}\bar{b}$. This naturally suggests the following group action: let $\mathbb{R}^\times$ denote the multiplicative group of nonzero real numbers. Define the action of $c \in \mathbb{R}^\times$ on $(a, b) \in \mathbb{R}^2$ by $c \cdot (a, b) \mapsto (ac, c^{-1}b)$. It is straightforward to verify that $\mathbb{R}^\times$ is a symmetry group of $f$. However, it does not fully capture the equivalence classes. Indeed, for $(a, b) \in \mathbb{R}^2$ with $ab \neq 0$, one has

$$[(a, b)] = \{(\bar{a}, \bar{b}) \in \mathbb{R}^2 \mid ab = \bar{a}\bar{b}\} = \{(ac, c^{-1}b) \mid c \in \mathbb{R}^\times\} = \mathbb{R}^\times(a, b). \tag{2}$$

In contrast, for $(a, b) \in \mathbb{R}^2$ with $ab = 0$, one obtains $[(a, b)] = \mathbb{R}^\times(1, 0) \sqcup \mathbb{R}^\times(0, 1) \sqcup \mathbb{R}^\times(0, 0)$. Thus, while $\mathbb{R}^\times$ almost completely describes the functional partition, it fails on the degenerate subset

of the parameter space where $ab = 0$. It is difficult to identify a larger natural group that extends the action to cover these exceptional cases.

*Second observation.* Classical group theory ensures that any partition of a set can be realized as the orbit decomposition of some group action. Accordingly, there always exists a group $G$ and an action of $G$ on $\Theta$ such that the $G$-orbits match the functional partition. However, constructing such a group typically requires explicit mappings, which are often intractable and impractical. In the context of parameterized models, where $\Theta$ is a finite-dimensional real space, it is natural to focus on group actions arising from standard operators such as addition, multiplication, or permutation.

These observations present a trade-off: the *tractability* of the group and its action versus the *descriptive capacity* of the functional partition, motivating the notion of maximality of symmetry groups.

**Maximal symmetry group.** The above observations lead to the following intuitive and informal description of a maximal symmetry group:

> *Under generic parameters, the symmetry group $G$ captures all functional equivalences,*
> *up to a sufficiently small exceptional set.*

In other words, let $\varepsilon$ denote a sufficiently small subset of $\Theta$, and consider the restricted domain $\Theta \setminus \varepsilon$. The group action of $G$ on $\Theta$ restricts naturally to $\Theta \setminus \varepsilon$. Then, for all $\theta, \bar{\theta} \in \Theta \setminus \varepsilon$ such that $f(\cdot; \theta) = f(\cdot; \bar{\theta})$, there exists $g \in G$ with $\bar{\theta} = g\theta$. Hence, although there may exist parameters in $\Theta$ for which $G$ does not capture FE, this exceptional set is negligible, and $G$ may still be regarded as characterizing FE of $\Theta$. The subset $\varepsilon$ is typically taken to coincide with the zero set of finitely many nonzero polynomials, that is, a proper real algebraic variety, consistent with prior work on FE in neural architectures (Hecht-Nielsen, 1990; Fefferman & Markel, 1993; Mai & Lampert, 2020).

**Definition 2.2** (Maximal symmetry group). A symmetry group $G$ is said to be *maximal* if there exists a proper real algebraic variety $\varepsilon \subsetneq \Theta$ such that, for all $\theta, \bar{\theta} \in \Theta \setminus \varepsilon$, whenever $f(\cdot; \theta) = f(\cdot; \bar{\theta})$, there exists $g \in G$ with $\bar{\theta} = g\theta$.

**Remark 2.3.** In the above observation on $f(\cdot; a, b)$, let $\varepsilon = \{(a, b) \in \mathbb{R}^2 : ab = 0\}$. Then $\varepsilon$ is a proper real algebraic variety, and the group $\mathbb{R}^\times$ serves as a maximal symmetry group of $f$.

In the next section, we demonstrate that this notion of maximality coincides with prior analyses of FE in feedforward and convolutional neural networks, as well as in multihead attention.

## 2.2 On the Role of Equivariance in Metanetworks

A *metanetwork* is a map $F : \Theta \rightarrow \mathcal{X}$ that takes as input the parameters of a model. Depending on the application, $F$ may return another element of $\Theta$ (as in network editing tasks) or a vector in $\mathbb{R}^d$ for some integer $d$ (as in prediction tasks). The fundamental objective is to determine whether the parameters of a model contain sufficient information to reveal properties of the function realized by the model itself. Since $F$ receives $\theta$ as input, it is natural to require that $F$ depend only on the underlying function represented by $\theta$, rather than on the particular parameterization. Equivalently, the input of $F$ should be the equivalence class $[\theta]$, as all elements of $[\theta]$ define the same function. It would be undesirable for $F(\theta)$ and $F(\bar{\theta})$ to produce incompatible outcomes whenever $[\theta] = [\bar{\theta}]$.

A principled approach to this requirement is to impose equivariance or invariance with respect to a symmetry group $G$. In particular, suppose $F : \Theta \rightarrow \Theta$ is $G$-equivariant. By definition, this means

$$F(g\theta) = gF(\theta), \quad \text{for all } g \in G, \ \theta \in \Theta. \tag{3}$$

Consequently, the equivalence classes are preserved in the sense that $[F(g\theta)] = [gF(\theta)] = [F(\theta)]$, thereby ensuring consistency across parameterizations that correspond to the same function. If $G$ is a maximal symmetry group of the underlying model, such equivariance is sufficient to guarantee that $F$ operates solely on the functional content of $\theta$. This observation underscores the importance of characterizing the maximal symmetry group–equivalently, of understanding FE–as a prerequisite for the systematic study of equivariant metanetworks.

## 3 Is Strict Equivariance Necessary for Metanetwork?

As discussed in Section 2.2, equivariance provides a principled mechanism for preserving the functional behavior of input networks. Nevertheless, equivariance should be regarded as a *sufficient*

condition for such preservation, rather than a necessary one. This naturally leads to the question:

*Is strict equivariance necessary for metanetworks?*

We now introduce a broader notion, namely quasi-equivariance. Throughout the remainder of the paper, let $G$ denote the maximal symmetry group, and let $F$ denote a metanetwork map.

**Quasi-equivariance.** We first address equivariance, deferring the discussion of invariance to a later stage. The requirement of functionality preservation for a map $F \colon \Theta \to \Theta$ can be stated as follows: for all $\bar{\theta} \in [\theta]$, one requires that $F(\bar{\theta}) \in [F(\theta)]$. By the maximality of $G$, the condition $\bar{\theta} \in [\theta]$ implies that there exists $g \in G$ such that $\bar{\theta} = g\theta$. The same holds for $F(\theta)$ and $F(\bar{\theta})$. Consequently, the above requirement can be reformulated, motivating the following definition.

**Definition 3.1** (Quasi-equivariance). A map $F \colon \Theta \to \Theta$ is said to be *G-quasi-equivariant* if, for all $g \in G$ and $\theta \in \Theta$, there exists $g' = g'(g, \theta) \in G$ such that $F(g\theta) = g'F(\theta)$.

The notation $g' = g'(g, \theta)$ emphasizes that $g'$ may depend on both $g$ and $\theta$. Figure 1 illustrates the quasi-equivariance property. By definition, every $G$-equivariant map is also $G$-quasi-equivariant. Moreover, $G$-quasi-equivariance ensures functionality preservation. Given the maximality of $G$ (Definition 2.2), it provides a *necessary and sufficient* condition for a map to preserve functionality. Indeed, for $\theta, \bar{\theta} \in \Theta$ such that $[\theta] = [\bar{\theta}]$, one has $\bar{\theta} = g\theta$ for some $g \in G$. Thus, $F(\bar{\theta}) = F(g\theta) = g'F(\theta)$ for some $g' \in G$. Therefore, $[F(\bar{\theta})] = [g'F(\theta)] = [F(\theta)]$.

**Remark 3.2.** A natural question is how to construct a map $F$ that satisfies the quasi-equivariant property. By Definition 3.1, one natural attempt is to first choose an arbitrary group-valued function $\alpha : G \times \Theta \to G$ and then solve for a map $F : \Theta \to \Theta$ satisfying $F(g\theta) = \alpha(g, \theta) F(\theta)$. However, for a general choice of $\alpha$, such an $F$ does not exist. Appendix A.1 provides the necessary conditions on $\alpha$ under which at least one corresponding $F$ can exist. Although this approach is theoretically motivated, it is not practical for constructing metanetworks. Therefore, in Section 4, we will present a more effective and implementable design for $F$.

**Invariance.** For invariance, introducing a quasi-version is unnecessary. Indeed, to ensure that a map $F \colon \Theta \to \mathcal{X}$ preserves functionality, it suffices to require $F(\bar{\theta}) = F(\theta)$, which is equivalent to $F(g\theta) = F(\theta)$. Hence, strict invariance is necessary.

**Properties.** In practice, equivariant and invariant metanetworks are constructed by stacking equivariant and invariant layers on top of one another, in the same manner as deep models are typically built. This construction relies on standard closure properties: the composition of two equivariant maps is equivariant, and the composition of an equivariant map with an invariant map is invariant. For quasi-equivariance, analogous properties hold, as stated in the following result.

**Proposition 3.3** (Composition). *Let $\varphi, \psi \colon \Theta \to \Theta$ be maps. Then:*

1. *If both $\varphi$ and $\psi$ are G-quasi-equivariant, then $\psi \circ \varphi$ is G-quasi-equivariant.*

2. *If $\varphi$ is G-quasi-equivariant and $\psi$ is G-invariant, then $\psi \circ \varphi$ is G-invariant.*

The assumption that $\varphi, \psi \colon \Theta \to \Theta$ is made for notational simplicity. In fact, the results remain valid if $\Theta$ is replaced by any domain on which the notions of $G$-quasi-equivariance and $G$-invariance are well-defined. The proof of Proposition 3.3 is provided in Appendix A.2.

**Remark 3.4** (Comparison with relaxed notions of equivariance in the literature). In Kaba & Ravanbakhsh (2023), the notion of *relaxed equivariance* is introduced. Given a group $G$ acting on $\mathcal{X}$ and $\mathcal{Y}$, a map $\varphi \colon \mathcal{X} \to \mathcal{Y}$ is said to satisfy relaxed equivariance if, for all $g \in G$ and $x \in \mathcal{X}$, there exists $g' \in gG_x$–with $G_x$ the stabilizer subgroup of $x$–such that $\varphi(gx) = g'\varphi(x)$. This definition is naturally subsumed under the broader notion of quasi-equivariance given in Definition 3.1. Other works, such as Wang et al. (2024), employ a related but distinct perspective, where relaxed equivariance is interpreted as an approximation to strict equivariance, namely $\varphi(gx) \approx g\varphi(x)$.

## 4 QUASI-EQUIVARIANT METANETWORKS

In this section, we establish a general framework for quasi-equivariant metanetworks and subsequently apply it to feedforward neural networks and multihead attention.

### 4.1 A General Framework for the Design of Quasi-Equivariant Metanetworks

Given the notation $f$, $\theta$, $\Theta$, and $G$ from the previous section, we focus on constructing $G$–quasi-equivariant networks. The invariant case is immediate, since it can be obtained by stacking an invariant layer on top of an equivariant backbone, as observed in Proposition 3.3.

A $G$–quasi-equivariant layer is defined as follows. Let $\alpha \colon \Theta \to G$ be a map into the group, and let $\beta \colon \Theta \to \Theta$ be an equivariant map. Define

$$F \colon \Theta \to \Theta, \qquad F(\theta) := \alpha(\theta)\beta(\theta). \tag{4}$$

By construction, $F$ is $G$-quasi-equivariant. In this framework, the design of $\beta$ follows directly from prior work on equivariant metanetworks. The central task is therefore to construct $\alpha$ so that it outputs group elements of $G$. This extension is motivated by the observation that strict equivariance is not necessary for metanetworks, and that enforcing it often yields sparse models due to the strong constraints imposed on the network weights. By introducing $\alpha$, we aim to relax these constraints, thereby improving both the expressivity and performance of metanetworks.

We now examine several representative cases of $G$, assuming $\alpha$ is continuous. In machine learning, continuity–and in practice differentiability–is essential for gradient-based optimization via backpropagation. The parameterized maps $f$ considered here will primarily be feedforward networks, convolutional neural networks, and multihead attention modules, as these are the predominant architectures in existing datasets of pretrained weights.

**Remark 4.1.** In the instances considered in the next part, the group $G$, although a group, can also be embedded into $\mathbb{R}^n$ for some $n$. In this setting, the group-valued map $\alpha : \Theta \to G$ may be regarded as continuous. We then recall the classical fact that the continuous image of a connected space is connected. Since $\Theta = \mathbb{R}^d$ is connected, the image $\alpha(\Theta)$ must also be connected. Consequently, if $G$ is discrete, $\alpha$ must be constant. This observation will be useful in the next part: when $G$ contains a discrete component (such as permutations), any continuous $\alpha$ cannot meaningfully vary over $\Theta$. Hence, the discrete part of $G$ can be ignored in the construction, and the focus is placed on the continuous component of $G$.

### 4.2 The Case of Feedforward and Convolutional Neural Networks

We primarily focus on the feedforward neural network. The convolutional counterpart can be treated in an analogous manner without loss of generality.

**Parameter space.** Consider a feedforward neural network $f$ with $L$ layers, having $n_i$ neurons in the $i^{\text{th}}$ layer and activation $\sigma$. Here, $n_0$ and $n_L$ are the input and output dimensions. The map $f$ is parameterized by $\theta = \{W_i, b_i\}_{i=1}^{L}$, where $W_i \in \mathbb{R}^{n_i \times n_{i-1}}$ and $b_i \in \mathbb{R}^{n_i}$. It is expressed as:

$$f(x; \theta) = f_L \circ \sigma \circ f_{L-1} \circ \sigma \circ \cdots \circ \sigma \circ f_1(x), \tag{5}$$

where $f_i \colon \mathbb{R}^{n_{i-1}} \to \mathbb{R}^{n_i}$ such that $x \mapsto W_i \cdot x + b_i$. The *parameter space* of $f$ is:

$$\Theta = \left(\mathbb{R}^{n_L \times n_{L-1}} \times \mathbb{R}^{n_L}\right) \times \cdots \times \left(\mathbb{R}^{n_2 \times n_1} \times \mathbb{R}^{n_2}\right) \times \left(\mathbb{R}^{n_1 \times n_0} \times \mathbb{R}^{n_1}\right) \tag{6}$$

**Maximal symmetry group.** We define a group action on $\Theta$ by monomial matrices. Let $n$ be a positive integer. A *monomial matrix* of size $n \times n$ is a matrix in which each row and each column contains exactly one nonzero entry. Denote $\mathcal{G}_n^{>0}$ as the sets of monomial matrices of size $n \times n$ with all non-zero entries positive. Now, define the group $G := \mathcal{G}_{n_{L-1}}^{>0} \times \ldots \times \mathcal{G}_{n_1}^{>0}$. Denote $g = (g_{L-1}, \ldots, g_1)$, where $g_i \in \mathcal{G}_{n_i}$, for elements of $G$. By convention, denote $g_L = I_{n_L}$ and $g_0 = I_{n_0}$, which are identity matrices. The *group action* of $G$ on $\Theta$ is defined by

$$g\theta := \{\bar{W}_i, \bar{b}_i\}_{i=1}^{L}, \text{ where } \bar{W}_i = g_i \cdot W_i \cdot g_{i-1}^{-1} \text{ and } \bar{b}_i = g_i \cdot b_i. \tag{7}$$

It is straightforward to check that $G$ forms a symmetry group for $f$. One expects $G$ to be maximal, however, for the general setting of $f$, it is an open question whether $G$ is maximal. Prior studies only proved $G$ to be maximal when restricted to a restricted setting. For instance, if $n_L \geqslant \ldots \geqslant n_2 \geqslant n_1 > n_0 = 1$, then $G$ is maximal (Mai & Lampert, 2020; Grigsby et al., 2023).

**Design of the map $\alpha$.** First, we decompose the group $G$ into factors $\mathcal{G}_{n_i}$ for $i \in [L-1]$, and construct maps $\Theta \to \mathcal{G}_{n_i}$ for each $i$. More generally, the goal is to define a map $\Theta \to \mathcal{G}_n$ for an

arbitrary positive integer $n$. To this end, we further analyze the structure of $\mathcal{G}_n$ by decomposing it as follows. Define $\mathcal{P}_n$ as the set of monomial matrices whose nonzero entries are all equal to 1, that is, the set of permutation matrices. Consider also the set of $n \times n$ diagonal matrices with positive diagonal entries, which is isomorphic to $\mathbb{R}^n_{>0}$, where $\mathbb{R}_{>0}$ denotes the multiplicative group of positive real numbers. Every monomial matrix in $\mathcal{G}_n$ can be expressed uniquely as the product of such a diagonal matrix and a permutation matrix, that is,

$$\mathcal{G}_n = \{DP \ : \ D \in \mathbb{R}^n_{>0} \text{ and } P \in \mathcal{P}_n\}. \tag{8}$$

Formally, $\mathcal{G}_n$ is isomorphic to the semidirect product $\mathcal{G}_n = \mathbb{R}^n_{>0} \rtimes \mathcal{P}_n$ (see Dummit & Foote (2004)). Thus, constructing $\alpha \colon \Theta \to \mathcal{G}_n$ reduces to specifying two maps: one into $\mathcal{P}_n$ and the other into $\mathbb{R}^n_{>0}$.

*The case of the group $\mathcal{P}_n$.* Since $\mathcal{P}_n$ is discrete, any continuous map $\alpha \colon \Theta \to \mathcal{P}_n$ must be constant.

*The case of the group $\mathbb{R}^n_{>0}$.* The group $\mathbb{R}^n_{>0}$ can be further decomposed into its coordinate factors, so that the construction of $\alpha \colon \Theta \to \mathbb{R}^n_{>0}$ reduces to specifying $n$ independent maps $\alpha_j \colon \Theta \to \mathbb{R}_{>0}$ for $j \in [n]$. To do this, the main idea is to construct the map from $\theta \in \Theta$ to a vector of size $n$, after which we take the sin of entries, scale it by a small $\epsilon > 0$ and add a unit vector $1_n$. The detailed implementation of $\alpha$ in practice is described in Appendix B.1.

**Remark 4.2** (Extension to CNN). For CNNs, the approach follows the same principle but is adapted for convolutional filters. Each bias vector $b_i$ retains the same dimensions as in MLPs, while the convolution filter $W_i \in \mathbb{R}^{n_i \times n_{i-1} \times w}$ (for 1D convolution) or $W_i \in \mathbb{R}^{n_i \times n_{i-1} \times c \times w}$ (for 2D convolution) contains additional spatial dimensions. To handle this, we apply the group action to both the filter's channel dimensions and spatial dimensions. Specifically, the filter $W_i$ is treated as having dimensions $n_i \times n_{i-1} \times (cw)$, where $c$ represents the number of input channels (or more generally, any extra spatial dimensions like spatial channels). This allows us to apply the quasi-equivariant layer to the filter in a manner similar to the MLP case, where we perform the scaling operation across the output channels and input channels.

**Remark 4.3** (On activation functions beyond ReLU). In the literature, FE has also been studied for feedforward neural networks with other activation functions $\sigma$. For instance, in the case of the tanh activation, a maximal symmetry group can be determined (see Chen et al. (1993); Fefferman & Markel (1993)). However, for most widely used activation functions, the maximal symmetry group is discrete, making the construction of $\alpha$ trivial. For this reason, our analysis focuses on ReLU.

### 4.3 THE CASE OF MULTIHEAD ATTENTION

**Parameter space.** Let $d$ denote the token dimension, $L$ the sequence length, and $h$ the number of heads, where all are positive integers. Define the space of token sequences as $\mathcal{S} \coloneqq \sqcup_{L=1}^\infty \mathbb{R}^{L \times d}$. For a fixed head dimension $d_h$, let $W_i^Q, W_i^K, W_i^V, W_i^O \in \mathbb{R}^{d \times d_h}$ for each $i \in [h]$, and set $\theta = (W_i^Q, W_i^K, W_i^V, W_i^O)_{i=1}^h$. Given an input sequence $\mathbf{x} = (x_1, \ldots, x_L)^\top \in \mathbb{R}^{L \times d} \subset \mathcal{S}$, the Multihead Attention (MHA) mechanism with $h$ heads is defined by

$$\text{MHA}\,(\mathbf{x} \colon \theta) = \sum_{i=1}^h \text{softmax}\left(\left(\mathbf{x}W_i^Q\right)\left(\mathbf{x}W_i^K\right)^\top\right) \cdot \left(\mathbf{x}W_i^V\right)\left(W_i^O\right)^\top. \tag{9}$$

Here, the softmax operator is applied row-wise to the similarity matrix $(\mathbf{x}W_i^Q)(\mathbf{x}W_i^K)^\top \in \mathbb{R}^{L \times L}$, producing the attention for $\mathbf{x}$. Each row of this matrix forms a probability distribution that determines the relative influence of all input tokens on a given output token. Typically, the head dimension is set to $d_h = d/h$. The parameter space of the MHA map is then defined as $\Theta \coloneqq \left(\mathbb{R}^{d \times d_h}\right)^{4h}$.

We denote by $\text{GL}(d_h)$ the general linear group of degree $d_h$, i.e., the set of all invertible $d_h \times d_h$ real matrices.

**Maximal symmetry group.** Define the following group $G \coloneqq S_h \times (\text{GL}(d_h) \times \text{GL}(d_h))^h$. This group is exactly the direct product of the permutation group $S_h$ with $h$ copies of $\text{GL}(d_h) \times \text{GL}(d_h)$. Each element $g \in G$ can be written as $g \coloneqq (\sigma, (U_i, V_i)_{i=1}^h)$, where $\sigma \in S_h$ and $U_i, V_i \in \text{GL}(d_h)$. The group $G$ acts naturally on the parameter space $\Theta$ as follows:

$$g\theta \coloneqq \left(W_{\sigma(i)}^Q \cdot U_i^\top, W_{\sigma(i)}^K \cdot U_i^{-1}, W_{\sigma(i)}^V \cdot V_i^\top, W_{\sigma(i)}^O \cdot V_i^{-1}\right)_{i=1}^h. \tag{10}$$

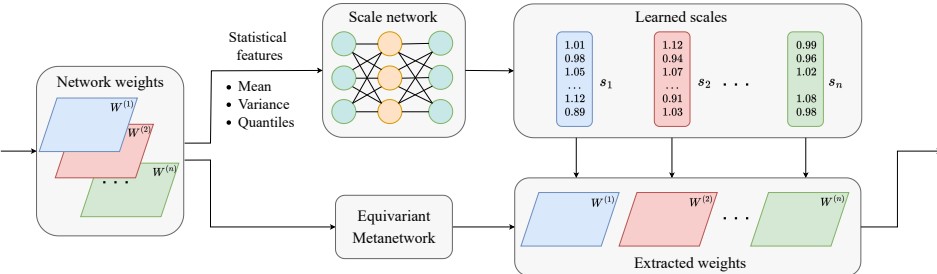

Figure 2: Illustration of the design of the quasi-equivariant layer. Statistical features are extracted from network weights and biases, then passed through a Scale network to learn the group action. This corresponds to the MLP case, where a scaling vector is learned for each layer's weights and biases. The learned scales are applied to the outputs of the equivariant layer, enhancing expressiveness while adding only minimal parameters.

It is evident that $G$ serves as a symmetry group of the MHA map. The reasoning is as follows: the general linear action cancels within the matrix multiplications, while the permutation action induced by $\sigma$ commutes with addition. Furthermore, $G$ is maximal, as formalized in the following result.

**Theorem 4.4** (See Tran et al. (2025b)). *Consider two* MHA *maps with $h$ heads, parameterized by $\theta = (W_i^Q, W_i^K, W_i^V, W_i^O)_{i=1}^h$ and $\bar{\theta} = (\bar{W}_i^Q, \bar{W}_i^K, \bar{W}_i^V, \bar{W}_i^O)_{i=1}^h$ in $\Theta$, respectively. Assume that*

1. *All matrices $W_i^Q, W_i^K, W_i^V, W_i^O$ and $\bar{W}_i^Q, \bar{W}_i^K, \bar{W}_i^V, \bar{W}_i^O$, for all feasible $i$, are of rank $d_h$.*

2. *From $\theta$, the matrices $\{W_i^Q(W_i^K)^\top\}_{i=1}^h$ are pairwise distinct. The same condition holds for $\bar{\theta}$.*

*If the two MHA maps are identical, there exists $g \in G$ such that $\bar{\theta} = g\theta$.*

**Remark 4.5.** Note that the conditions on $\theta$ and $\bar{\theta}$ in Theorem 4.4 can both be expressed as the vanishing of finitely many nonzero polynomials. This corresponds precisely to the real algebraic variety $\varepsilon$ introduced in Definition 2.2 of maximal symmetry groups.

**Design of the map $\alpha$.** In analogy with the feedforward case, we restrict the construction of $\alpha\colon \Theta \to G$ to the design of a map $\Theta \to \mathrm{GL}(n)$ for a general positive integer $n$. The idea is as follows: first, reshape $\theta \in \Theta$ into an $n \times n$ matrix via a feedforward network $\gamma$. Next, apply the entrywise sine function, scale the result by a small $\epsilon > 0$, and finally add the identity matrix $I_n$, i.e. $\theta \longmapsto \sin(\gamma(\theta)) \cdot \epsilon + I_n$. By continuity, and since the range of the sine function is $[-1, 1]$, there exists a sufficiently small $\epsilon > 0$ such that the resulting matrix is invertible. The detailed implementation of $\alpha$ in practice is described in Appendix B.1.

**Remark 4.6.** An alternative approach to constructing a map $\Theta \to \mathrm{GL}(n)$ is to use the matrix exponential $\exp\colon \mathbb{R}^{n \times n} \to \mathrm{GL}(n)$. However, both in theory and in practice, this approach tends to be slow and numerically unstable. Our experimental trials confirmed these issues, and we therefore did not pursue this direction further.

## 5 EXPERIMENTS

In this section, we integrate the quasi-equivariant layer with existing metanetworks: Monomial-NFN for MLP/CNNs and Transformer-NFN for Transformers. The overall layer is illustrated in Figure 2. We provide detailed implementation of the MLP network for each case (MLP/CNN or Transformers) in Appendix B.1. We evaluate these models on three tasks: predicting CNN generalization from weights, classifying image INRs, and predicting Transformer generalization from parameters. For each task, we compare against the original baseline and an expanded version with more parameters, ensuring fair comparison with our method. Our aim is twofold: the quasi-equivariant layer improves performance efficiently with minimal parameter increase, and it preserves performance under group action transformations. Results are averaged over five runs; hyperparameters and parameter counts are detailed in Appendix B.

### 5.1 PREDICTING CNN GENERALIZATION

**Experiment Setup.** We aim to predict pretrained CNN generalization using only their weights, without test data. Experiments use the Small CNN Zoo dataset (Unterthiner et al., 2020), containing

Table 1: Performance prediction of CNNs on the RELU subset of Small CNN Zoo with varying scale augmentations. The metric used is Kendall's $\tau$. Uncertainties indicate the standard error across 5 runs.

| | | Augment settings | | | |
|---|---|---|---|---|---|
| | No augment | $\mathcal{U}[1, 10^1]$ | $\mathcal{U}[1, 10^2]$ | $\mathcal{U}[1, 10^3]$ | $\mathcal{U}[1, 10^4]$ |
| STATNet (Unterthiner et al., 2020) | $0.915 \pm 0.002$ | $0.894 \pm 0.0001$ | $0.853 \pm 0.007$ | $0.523 \pm 0.02$ | $0.516 \pm 0.001$ |
| NP (Zhou et al., 2024a) | $0.920 \pm 0.003$ | $0.900 \pm 0.002$ | $0.898 \pm 0.003$ | $0.884 \pm 0.002$ | $0.884 \pm 0.002$ |
| HNP (Zhou et al., 2024a) | $\mathbf{0.926 \pm 0.003}$ | $0.913 \pm 0.001$ | $0.903 \pm 0.003$ | $0.891 \pm 0.003$ | $0.601 \pm 0.02$ |
| Graph-NN (Kofinas et al., 2024) | $0.897 \pm 0.002$ | $0.892 \pm 0.003$ | $0.885 \pm 0.002$ | $0.858 \pm 0.003$ | $0.851 \pm 0.002$ |
| Monomial-NFN (Tran et al., 2024a) | $0.922 \pm 0.001$ | $\underline{0.920 \pm 0.001}$ | $0.919 \pm 0.001$ | $\underline{0.920 \pm 0.002}$ | $\underline{0.920 \pm 0.001}$ |
| Monomial-NFN large ($\mathbf{68.65\%}$ params ++) | $\underline{0.923 \pm 0.001}$ | $\underline{0.920 \pm 0.001}$ | $\underline{0.920 \pm 0.002}$ | $0.919 \pm 0.001$ | $\underline{0.920 \pm 0.001}$ |
| Monomial-NFN Quasi (ours) ($\mathbf{3.89\%}$ params ++) | $\mathbf{0.926 \pm 0.002}$ | $\mathbf{0.924 \pm 0.002}$ | $\mathbf{0.924 \pm 0.002}$ | $\mathbf{0.923 \pm 0.001}$ | $\mathbf{0.924 \pm 0.002}$ |

Table 2: Classification train and test accuracies (%) for implicit neural representations of MNIST, FashionMNIST, and CIFAR-10. Uncertainties indicate standard error over 5 runs.

| | MNIST | CIFAR-10 | FashionMNIST |
|---|---|---|---|
| MLP | $10.62 \pm 0.54$ | $10.48 \pm 0.74$ | $9.95 \pm 0.36$ |
| NP (Zhou et al., 2024a) | $\underline{69.82 \pm 0.42}$ | $33.74 \pm 0.26$ | $58.21 \pm 0.31$ |
| HNP (Zhou et al., 2024a) | $66.02 \pm 0.51$ | $31.61 \pm 0.22$ | $57.43 \pm 0.46$ |
| Monomial-NFN (Tran et al., 2024a) | $68.43 \pm 0.51$ | $34.23 \pm 0.33$ | $61.15 \pm 0.55$ |
| Monomial-NFN tuned ($\approx \mathbf{3\%}$ params++) | $68.87 \pm 0.42$ | $\underline{34.26 \pm 0.28}$ | $\underline{61.44 \pm 0.35}$ |
| Monomial-NFN Quasi (ours) ($\approx \mathbf{3\%}$ params ++) | $\mathbf{70.21 \pm 0.34}$ | $\mathbf{35.32 \pm 0.56}$ | $\mathbf{62.11 \pm 0.27}$ |

CNNs trained with varying hyperparameters and activations. Following Tran et al. (2024a), we analyze the ReLU subset, where models follow the group action $\mathcal{M}_n^{>0}$. Robustness to group-action transformations is evaluated by augmenting the dataset with variants from diagonal matrices $\mathcal{D}_{n,ii}^{>0} \sim \mathcal{U}[1, 10^i]$ for $i \in \{1, 2, 3, 4\}$ and random permutation matrices $\mathcal{P}_n$. Prediction is measured with Kendall's $\tau$ rank correlation (Kendall, 1938), which quantifies agreement between predicted and true accuracy rankings. Our approach extends Monomial-NFN (Tran et al., 2024a), denoted Monomial-NFN Quasi, and is compared with STATNN (Unterthiner et al., 2020), NP, HNP (Zhou et al., 2024a), and Graph-NN (Kofinas et al., 2024). For Monomial-NFN, we test both the original and an enlarged, carefully tuned variant for fair comparison.

**Results.** Table 1 reports model performance. Scaling Monomial-NFN (+68.65% parameters) gives minor gains, whereas Monomial-NFN Quasi shows notable improvement with only +3.89% parameters. On the original subset, where Monomial-NFN lags behind HNP, the Quasi layer matches its performance. This holds under augmentation, showing that small parameter increases via the Quasi-equivariant layer can enhance expressiveness and flexibility considerably.

## 5.2 CLASSIFYING IMPLICIT NEURAL REPRESENTATIONS OF IMAGES

**Experiment Setup.** This experiment focuses on classifying the source class of pretrained Implicit Neural Representation (INR) weights. Following the setup in (Tran et al., 2024a), we use three INR weight datasets introduced in (Zhou et al., 2024a), each corresponding to a different image dataset: CIFAR-10 (Krizhevsky & Hinton, 2009), FashionMNIST (Xiao et al., 2017), and MNIST (LeCun & Cortes, 2005). In these datasets, each INR is trained to represent a single image, encoding image structure by mapping pixel coordinates $(x, y)$ to pixel color values. CIFAR-10 images are encoded as 3-channel RGB outputs, while MNIST and FashionMNIST are represented with a single grayscale channel. Since excessively increasing parameters in Monomial-NFN leads to overfitting in this setting, we introduce a variant, Monomial-NFN tuned, which is carefully adjusted to match the parameter count of Monomial-NFN Quasi. This ensures a fair comparison between the two models.

**Results.** Table 2 shows the performance of all models on INR classification. The tuned Monomial-NFN, which adds 3% more parameters, yields only minor improvement. In contrast, adding the proposed quasi layer with the same parameter increase allows Monomial-NFN Quasi to outperform NP on the MNIST task and achieve the best results across all three datasets. The performance gap between Monomial-NFN Quasi and Monomial-NFN is about 1% on CIFAR-10 and FashionMNIST, and 1.78% on MNIST. These results demonstrate the consistency of the proposed method.

Table 3: Performance measured by Kendall's $\tau$ of all models on MNIST- and AGNews-Transformers datasets. Uncertainties indicate standard error over 5 runs.

| | Accuracy threshold | | | | |
|---|---|---|---|---|---|
| | No threshold | 20% | 40% | 60% | 80% |
| *MNIST-Transformers* | | | | | |
| MLP | $0.866 \pm 0.002$ | $0.873 \pm 0.001$ | $0.874 \pm 0.003$ | $0.874 \pm 0.006$ | $0.873 \pm 0.007$ |
| STATNN (Unterthiner et al., 2020) | $0.881 \pm 0.001$ | $0.872 \pm 0.001$ | $0.868 \pm 0.001$ | $0.860 \pm 0.001$ | $0.856 \pm 0.001$ |
| XGBoost (Chen & Guestrin, 2016) | $0.860 \pm 0.002$ | $0.839 \pm 0.004$ | $0.869 \pm 0.003$ | $0.846 \pm 0.001$ | $0.884 \pm 0.001$ |
| LightGBM (Ke et al., 2017) | $0.858 \pm 0.002$ | $0.835 \pm 0.001$ | $0.847 \pm 0.001$ | $0.822 \pm 0.001$ | $0.830 \pm 0.001$ |
| Random Forest (Breiman, 2001) | $0.772 \pm 0.002$ | $0.758 \pm 0.004$ | $0.769 \pm 0.001$ | $0.752 \pm 0.001$ | $0.759 \pm 0.001$ |
| Transformer-NFN (Tran et al., 2025b) | $0.905 \pm 0.002$ | $0.899 \pm 0.001$ | $0.895 \pm 0.001$ | $0.895 \pm 0.002$ | $0.888 \pm 0.002$ |
| Transformer-NFN large (57.66% params ++) | $\underline{0.907 \pm 0.001}$ | $\underline{0.904 \pm 0.002}$ | $\underline{0.897 \pm 0.002}$ | $\underline{0.897 \pm 0.002}$ | $\underline{0.890 \pm 0.001}$ |
| Transformer-NFN Quasi (Ours) (4.54% params ++) | $\mathbf{0.911 \pm 0.001}$ | $\mathbf{0.905 \pm 0.001}$ | $\mathbf{0.898 \pm 0.002}$ | $\mathbf{0.897 \pm 0.001}$ | $\mathbf{0.892 \pm 0.001}$ |
| *AGNews-Transformers* | | | | | |
| MLP | $0.879 \pm 0.006$ | $0.875 \pm 0.001$ | $0.841 \pm 0.012$ | $0.842 \pm 0.001$ | $0.862 \pm 0.006$ |
| STATNN (Unterthiner et al., 2020) | $0.841 \pm 0.002$ | $0.839 \pm 0.003$ | $0.812 \pm 0.003$ | $0.813 \pm 0.001$ | $0.812 \pm 0.001$ |
| XGBoost (Chen & Guestrin, 2016) | $0.859 \pm 0.001$ | $0.852 \pm 0.002$ | $0.872 \pm 0.002$ | $0.874 \pm 0.001$ | $0.872 \pm 0.001$ |
| LightGBM (Ke et al., 2017) | $0.835 \pm 0.001$ | $0.845 \pm 0.001$ | $0.837 \pm 0.001$ | $0.835 \pm 0.001$ | $0.820 \pm 0.001$ |
| Random Forest (Breiman, 2001) | $0.774 \pm 0.003$ | $0.801 \pm 0.001$ | $0.797 \pm 0.001$ | $0.798 \pm 0.002$ | $0.773 \pm 0.001$ |
| Transformer-NFN (Tran et al., 2025b) | $0.910 \pm 0.001$ | $0.908 \pm 0.001$ | $0.897 \pm 0.001$ | $0.896 \pm 0.001$ | $0.890 \pm 0.001$ |
| Transformer-NFN large (59.38% params ++) | $\underline{0.913 \pm 0.001}$ | $\underline{0.910 \pm 0.002}$ | $\underline{0.898 \pm 0.002}$ | $\underline{0.898 \pm 0.001}$ | $\underline{0.893 \pm 0.002}$ |
| Transformer-NFN Quasi (Ours) (5.27% params ++) | $\mathbf{0.914 \pm 0.001}$ | $\mathbf{0.913 \pm 0.002}$ | $\mathbf{0.901 \pm 0.001}$ | $\mathbf{0.903 \pm 0.002}$ | $\mathbf{0.896 \pm 0.001}$ |

## 5.3 PREDICTING TRANSFORMERS GENERALIZATION

**Experiment Setup.** In this task, we predict the accuracies of pretrained Transformer checkpoints, aiming to test whether metanetworks capture structural patterns in Transformer weights. Following (Tran et al., 2025b), we integrate our quasi-layer into Transformer-NFN and conduct evaluations on two datasets: MNIST-Transformers, built from models trained for MNIST image classification, and AGNews-Transformers, derived from models trained for AGNews text classification. Performance is assessed using Kendall's $\tau$. To probe model performance under varying difficulty levels, we evaluate not only on the full dataset but also on four subsets defined by minimum accuracy thresholds of 20%, 40%, 60%, and 80%. Since most pretrained models in these datasets achieve high accuracy, maintaining strong Kendall's $\tau$ becomes more difficult as the threshold increases.

**Results.** Table 3 presents the results on MNIST-Transformer and AGNews-Transformer. In both benchmarks, scaling up Transformer-NFN to a larger version (with up to 59.38% more parameters) can improve Kendall's $\tau$, but adding the quasi-equivariant layer achieves even greater gains with far fewer extra parameters (only up to 5.27%). The improvement holds across all accuracy thresholds, demonstrating both the efficiency and effectiveness of our approach.

## 6 CONCLUSIONS, LIMITATIONS, AND FUTURE DIRECTIONS

This paper introduces quasi-equivariant metanetworks, a framework that relaxes strict equivariance to balance symmetry preservation with representational flexibility. By analyzing parameter spaces, their symmetry groups, and maximality, we establish a theoretical connection to functional equivalence and formalize quasi-equivariance as a principled extension of strict equivariance. We demonstrate applicability of the framework to feedforward networks and multihead attention, and validate its effectiveness across multiple metanetwork benchmarks, achieving substantial performance gains with minimal additional parameters. A current limitation of our work is that quasi-equivariance has so far been applied primarily to metanetworks with linear architectures. Extending this framework to more complex structures, such as graph-based metanetworks, remains unexplored due to the diversity and rarity of such architectures. Moreover, the quasi-equivariant design could be beneficial in various other fields, such as computational chemistry, physics, and materials science, where symmetries may only approximately hold and greater modeling flexibility is required. We view these as promising directions for future research.

ACKNOWLEDGMENTS

This research / project is supported by the National Research Foundation Singapore under the AI Singapore Programme (AISG Award No: AISG2-TC-2023-012-SGIL). This research / project is supported by the Ministry of Education, Singapore, under the Academic Research Fund Tier 1 (FY2023) (A-8002040-00-00, A-8002039-00-00). This research / project is also supported by the NUS Presidential Young Professorship Award (A-0009807-01-00) and the NUS Artificial Intelligence Institute–Seed Funding (A-8003062-00-00).

**Ethics Statement.** Considering the scope and focus of this work, we do not anticipate any adverse societal or ethical consequences.

**Reproducibility Statement.** The source code for all experiments is included in the paper's supplementary materials. Detailed descriptions of our experimental setup can be found in Section 5 and Appendix B. All datasets used are publicly accessible via an anonymous link provided in the README of the supplementary materials.

**LLM Usage.** In this work, we use large language models (LLMs) solely as a tool to assist and refine the presentation of our ideas. The LLM was employed only to improve clarity, grammar, and overall readability, without influencing the scientific content, methodology, or experimental results. All technical contributions, analyses, and conclusions in the paper are entirely the authors' original work.

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

TABLE OF NOTATION

| | |
|---|---|
| $\Theta$ | Parameter space of the network |
| $W_i$ | Weight of feed forward neural network in layer $i$ |
| $b_i$ | Bias of feed forward neural network in layer $i$ |
| $\mathcal{G}_n$ | Monomial matrix of size $n$ |
| $h$ | Number of head of Attention module |
| $d$ | Hidden dimension of the model |
| $d_h$ | Hidden dimension of a head in the model |
| $D_k$ | Dimension of key/query vector in Attention module |
| $W_i^Q$ | Weight of query matrix of head $i$ |
| $W_i^K$ | Weight of key matrix of head $i$ |
| $W_i^V$ | Weight of value matrix of head $i$ |
| $W_i^O$ | Weight of out projection matrix of head $i$ |
| $G$ | Symmetric group of the weight space |
| $\sigma()$ | Relu activation |
| $S_h$ | Head permutation group action in Attention module |
| $E()$ | Equivariant layer |
| $I()$ | Invariant layer |
| $\mathbb{R}^d$ | $d$-dimensional Euclidean space |
| $\langle \cdot, \cdot \rangle$ | standard dot product |
| $\sqcup$ | disjoint union |
| $g$ | element of group |
| $\mathrm{GL}(d_h)$ | General linear group of invertible $d_h \times d_h$ matrices over $\mathbb{R}$ |
| $\alpha()$ | Quasi-equivariant map |
| $\beta()$ | Equivariant map |

# Supplement to "Quasi-Equivariant Metanetworks"

**Table of Contents**

## A  QUASI-EQUIVARIANT METANETWORKS

**Invariance and Equivariance in Machine Learning.**  Equivariant neural networks (Cohen & Welling, 2016) incorporate task-specific symmetries directly into their architectures, leading to improved generalization and greater sample efficiency. They have achieved strong empirical success across diverse domains, including trajectory prediction (Walters et al., 2020), robotics (Simeonov et al., 2022), graph representation learning (Satorras et al., 2021; Tran et al., 2024b), and Optimal Transport–based methods (Pham et al., 2026; Tran et al., 2026a;b; 2024c; 2025e;d;a;c). Leveraging equivariance consistently enhances performance, data efficiency, and robustness under distribution shifts.

**Quasi-Equivariance.**  In the main paper, we introduced the concept of quasi-equivariance and discussed its key insights. Here, we establish formal theoretical guarantees and provide rigorous proofs for the results related to this notion.

### A.1  ON THE WELL-DEFINEDNESS OF THE QUASI-EQUIVARIANCE PROPERTY

We now examine the well-definedness of the quasi-equivariance property, as stated in Remark 3.2. Although the construction of quasi-equivariant maps discussed in this section is not used in our implementation due to its inefficiency, we include the following analysis for completeness and for potential future work, where one may wish to construct quasi-equivariant maps using this approach.

**Setup.**  Let a group $G$ act on a set $\Theta$. Let a (possibly the same) group $H$ act on a set $\mathcal{X}$. A map $F : \Theta \to \mathcal{X}$ is called $\alpha$–*quasi–equivariant* if there exists

$$\alpha : G \times \Theta \to H \tag{11}$$

such that for all $g \in G$ and $\theta \in \Theta$,

$$F(g\theta) = \alpha(g, \theta) \cdot F(\theta). \tag{12}$$

When $H = G$ and the group action of $H$ on $\mathcal{X}$ is the given group action of $G$, this reduces to our original formulation on quasi-equivariance.

**Well-definedness across different representatives.**  The relation $g_1\theta_1 = g_2\theta_2$ gives two representations of the same point in $\Theta$. Thus Equation (13) must produce the same value of $F$. A necessary and sufficient condition on $\alpha$ is presented as follows.

**Proposition A.1** (Normalized 1-cocycle condition)**.** *The following statements are equivalent:*

1. *For every choice of a section $S \subset \Theta$ meeting each $G$-orbit exactly once, and for every seed map $s : S \to \mathcal{X}$ satisfying the stabilizer constraint (see Proposition A.2), there exists a unique map $F : \Theta \to \mathcal{X}$ obeying the quasi–equivariance relation*

$$F(g\theta) = \alpha(g, \theta) \cdot F(\theta), \quad \text{for all } g \in G, \theta \in \Theta, \tag{13}$$

   *such that $F$ is independent of the choice of representative of a point in $\Theta$.*

2. *The function $\alpha : G \times \Theta \to H$ satisfies, for all $g_1, g_2 \in G$ and $\theta \in \Theta$,*

$$\alpha(e, \theta) = e_H, \tag{14}$$
$$\alpha(g_1 g_2, \theta) = \alpha(g_1, \, g_2\theta) \, \alpha(g_2, \theta). \tag{15}$$

*Proof.* The proof proceeds as follows.

*Necessity.* Fix $g_1, g_2, \theta$. Using Equation (13) twice,

$$F(g_1 g_2 \theta) = \alpha(g_1, \, g_2\theta) \cdot F(g_2\theta) = \alpha(g_1, \, g_2\theta) \, \alpha(g_2, \theta) \cdot F(\theta). \tag{16}$$

On the other hand, applying Equation (13) once with $g_1 g_2$ gives

$$F(g_1 g_2 \theta) = \alpha(g_1 g_2, \theta) \cdot F(\theta), \tag{17}$$

so Equation (15) of 1-cocycle follows. Setting $g = e$ in Equation (13) yields Equation (14) of Normalization.

*Sufficiency.* Given any representative $g\theta_0$ of a point in the orbit of $\theta_0$, define $F(g\theta_0)$ by Equation (13). If $g_1\theta_0 = g_2\theta_0$, then $g_2^{-1}g_1 \in G_{\theta_0}$ and the cocycle identity implies the two definitions coincide provided the stabilizer constraint in Proposition A.2 holds. $\qquad\square$

**Stabilizers and orbit descent.** Denote $G_\theta = \{h \in G : h\theta = \theta\}$. The values of $\alpha$ on $G_\theta$ control whether $F$ is well-defined from orbit data alone.

**Proposition A.2** (Stabilizer constraints)**.** *Assume Equations (14) and (15). Then for each $\theta$ and $h \in G_\theta$, one has*

$$F(\theta) = F(h\theta) = \alpha(h, \theta) \cdot F(\theta). \tag{18}$$

*Hence $F(\theta)$ must lie in the fixed-point set*

$$\text{Fix}_{\mathcal{X}}\big(\alpha(G_\theta, \theta)\big) = \{x \in \mathcal{X} : \alpha(h, \theta) \cdot x = x \text{ for all } h \in G_\theta\}. \tag{19}$$

*In particular:*

- *If $F$ is required to be definable independently of any additional constraints on its values (i.e., to descend to $\Theta/G$ without restricting the image), then a necessary and sufficient condition is*

$$\alpha(h, \theta) = e_H \quad \text{for all } h \in G_\theta, \ \theta \in \Theta. \tag{20}$$

- *More generally, $F(\theta)$ is allowed to live in the moving fixed-point set above; then triviality on stabilizers is not required, but the image of $F$ is constrained.*

**Gauge/coboundary equivalence.** Two quasi–equivariant structures related by a change of variables in $\mathcal{X}$ are equivalent.

**Proposition A.3** (Gauge transform and trivial class)**.** *Let $\beta : \Theta \to H$. Define*

$$\alpha^\beta(g, \theta) := \beta(g\theta)\alpha(g, \theta)\,\beta(\theta)^{-1}, \qquad F^\beta(\theta) := \beta(\theta) \cdot F(\theta). \tag{21}$$

*Then $F$ satisfies Equation (13) with $\alpha$ if and only if $F^\beta$ satisfies Equation (13) with $\alpha^\beta$. In particular, if $\alpha$ is a coboundary, i.e.*

$$\alpha(g, \theta) = \beta(g\theta)\,\beta(\theta)^{-1}, \tag{22}$$

*then with $F'(\theta) := \beta(\theta)^{-1} \cdot F(\theta)$ one has the strict equivariance:*

$$F'(g\theta) = F'(\theta) \qquad\qquad \text{if } H \text{ acts trivially,} \quad \text{or} \tag{23}$$
$$F'(g\theta) = g \cdot F'(\theta) \qquad\qquad \text{if } \beta \text{ is valued in } G \text{ and } H = G. \tag{24}$$

*Thus, normalized cocycles modulo coboundaries classify quasi–equivariant structures up to gauge (a $H^1$ of the transformation groupoid $G \ltimes \Theta$ with coefficients in $H$).*

**Regularity (topological/smooth settings).** If $\Theta, \mathcal{X}$ are topological (or smooth) spaces and the group actions are continuous (or smooth), then to ensure $F$ is continuous (or smooth) whenever the seed $s$ is, one additionally asks $\alpha(\cdot, \cdot)$ to be continuous (or smooth) and the group actions to be continuous (or smooth). The results above remain valid verbatim.

**Conclusion.** We have the following observations.

1. **Strict equivariance.** If $\alpha(g, \theta) \equiv g$ (and $H = G$ with the given action), then Equation (15) is automatic and we recover the standard $G$–equivariance $F(g\theta) = g \cdot F(\theta)$.

2. **Homomorphic twist.** If $\alpha(g, \theta) = \psi(g)$ for a homomorphism $\psi : G \to H$, then Equation (15) holds. To descend to orbits without image constraints one needs $\psi(h) = e_H$ for all $h \in G_\theta$ and all $\theta$ (i.e., $\psi$ trivial on all stabilizers).

3. **Coboundary (gauge) case.** If $\alpha(g, \theta) = \beta(g\theta)\beta(\theta)^{-1}$, one can *gauge* to a strictly equivariant $F'$ as in Proposition A.3.

To make sense of the quasi–equivariance relation $F(g\theta) = \alpha(g, \theta) \cdot F(\theta)$ independently of how a point of $\Theta$ is represented, the essential structural requirement is that $\alpha$ be a *normalized* 1-*cocycle* on the transformation groupoid $G \ltimes \Theta$ with values in $H$ (Equations (14) and 15). Descent to the orbit space without restricting the image of $F$ additionally demands *triviality on stabilizers*. Up to gauge, quasi–equivariant structures are classified by the corresponding first cohomology set; coboundaries are precisely those that can be turned into strict equivariance by a change of variables.

**Remark A.4.** The three conclusions regarding the conditions on $\alpha$ serve as necessary requirements for the existence of a map $F$ satisfying

$$F(g\theta) = \alpha(g, \theta)F(\theta). \tag{25}$$

## A.2 PROOF OF PROPOSITION 3.3

In this section, we provide the proof for Proposition 3.3.

*Proof of Proposition 3.3.* We provide a proof of each part of the proposition.

*Part 1.* Assume $\varphi$ and $\psi$ are $G$-quasi-equivariant.

$$\forall h \in G, \ \forall \theta \in \Theta, \ \exists h_1 \in G : \ \varphi(h\theta) = h_1\varphi(\theta). \tag{26}$$

Then

$$\forall h_1 \in G, \ \forall \vartheta \in \Theta, \ \exists h_2 \in G : \ \psi(h_1\vartheta) = h_2\psi(\vartheta). \tag{27}$$

Taking $\vartheta = \varphi(\theta)$,

$$\forall h \in G, \ \forall \theta \in \Theta, \ \exists h_2 \in G : \ \psi(\varphi(h\theta)) = \psi(h_1\varphi(\theta)) = h_2\psi(\varphi(\theta)). \tag{28}$$

Hence

$$\forall h \in G, \ \forall \theta \in \Theta, \ \exists h_2 \in G : \ (\psi \circ \varphi)(h\theta) = h_2(\psi \circ \varphi)(\theta), \tag{29}$$

so $\psi \circ \varphi$ is $G$-quasi-equivariant.

*Part 2.* Assume $\varphi$ is $G$-quasi-equivariant and $\psi$ is $G$-invariant:

$$\forall h \in G, \ \forall \theta \in \Theta, \ \exists h_1 \in G : \ \varphi(h\theta) = h_1\varphi(\theta), \tag{30}$$

and

$$\forall k \in G, \ \forall \vartheta \in \Theta : \ \psi(k\vartheta) = \psi(\vartheta). \tag{31}$$

Taking $k = h_1$ and $\vartheta = \varphi(\theta)$,

$$\forall h \in G, \ \forall \theta \in \Theta : \ \psi(\varphi(h\theta)) = \psi(h_1\varphi(\theta)) = \psi(\varphi(\theta)). \tag{32}$$

Therefore

$$\forall h \in G, \ \forall \theta \in \Theta : \ (\psi \circ \varphi)(h\theta) = (\psi \circ \varphi)(\theta), \tag{33}$$

so $\psi \circ \varphi$ is $G$-invariant.

The proof is complete. $\square$

# B   ADDITIONAL DETAILS OF EXPERIMENTS

## B.1   DETAILS ON GROUP ACTION LEARNING

We provide more details on building the process to learn group actions for two cases: MLP/CNN and Transformers.

Our quasi-equivariant layer is designed in two stages: feature selection and group action learning. The goal is to learn a group action ($\alpha$) from network weights and biases, and then apply it to the outputs of existing equivariant layers.

- **Feature Selection:** A straightforward approach is to flatten and concatenate weights and biases, but this introduces excessive parameters and becomes infeasible for large networks. To address this, inspired by STATNET (Unterthiner et al., 2020), we instead compute statistical features: mean, variance, and five quantiles (0, 0.25, 0.5, 0.75, 1) of weights and biases. These features are concatenated into a compact representation that scales consistently with network size while retaining essential information.

- **Group Action Learning:** We adopt a MLP to model the group action, tailored to the underlying MLP or Transformer weight space. To encourage stability, the action is learned around the identity. Inspired by Fourier analysis, where sine functions form basic signal components, we introduce a structured noise mechanism: $\tilde{a} = \sin(W_{\text{scale}}x + b_{\text{scale}}) \cdot \epsilon + \{1_n, I_n\}$, where $W_{\text{scale}}$ and $b_{\text{scale}}$ are parameters of MLP, and $\epsilon$ is a small learnable factor. This generates mild oscillations centered at unity, preserving the base equivariant behavior while providing flexibility for improved learning.

**Implementation of Quasi-Equivariant Layer for MLP/CNN weights.**   In MLPs and CNNs, our goal is to learn a positive scale vector for each layer. The group action is applied to the output dimensions of the weights and biases by scaling all neurons in the output dimension, while correspondingly scaling the input dimension of the subsequent layer with the reciprocal value of the scale.

To construct this, we first extract the weights and biases from the input network. For each layer, we compute seven statistical features: the mean, variance, and five quantiles (0, 0.25, 0.5, 0.75, 1). Features from weights and biases are concatenated, yielding a 14-dimensional representation per layer. Aggregating across all layers produces a feature vector of size $14 \cdot L$, where $L$ is the total number of layers in the network.

The group action for each layer is parameterized as a positive scale vector of size $N_{out}$, where $N_{out}$ denotes the layer's output dimension. We learn this vector using a Gated-MLP with hidden dimension 32, assigning a separate network to each layer. To ensure stability and positivity, we incorporate the structured noise mechanism described above. The resulting scale is then applied to the final equivariant layer of the metanetwork (specifically, Monomial-NFN).

**Implementation of Quasi-Equivariant Layer for Transformer weights.**   In Transformers, our objective is to learn two invertible matrices, $M$ and $N$, for each layer, representing the GL group action applied to $W_q, W_k, W_v$, and $W_o$. Specifically, $M$ is applied to the query and key matrices, yielding $W_q M^T$ and $W_k M^{-1}$. This ensures that the attention score remains unchanged, since

$$(W_q M^T)(W_k M^{-1})^T = W_q W_k^T.$$

Similarly, $N$ is applied to the value and output projection matrices, transforming them into $W_v N$ and $N^{-1} W_o$.

To construct these transformations, we extract all weights and biases from the input network: $W_q, W_k, W_v, W_o, W_A, b_A, W_B, b_B$. For each layer, we compute seven statistical features—the mean, variance, and five quantiles (0, 0.25, 0.5, 0.75, 1). Concatenating features from all weights and biases gives a 56-dimensional representation per layer. Aggregating across $L$ layers results in a feature vector of size $56 \cdot L$.

The group action for each layer is parameterized as two invertible matrices of shape $(D, D)$, where $D$ denotes the hidden dimension in attention, and each attention head has its own pair $(M, N)$. To learn these matrices, we employ an MLP with hidden dimension 32, assigning one network per layer. The MLP maps the feature vector to $n_h \cdot D \cdot D$, which is then reshaped into $(n_h, D, D)$. To guarantee stability and numerical soundness, we apply the structured noise mechanism described above. This procedure produces matrices that are invertible almost everywhere, allowing stable training. The learned transformations are finally applied to the last equivariant layer of the metanetwork (specifically, Transformer-NFN).

**On the design of quasi-equivariant layer.** In the MLP/CNN case, the map $\alpha$ consists of two parts: a constant map for the group $\mathcal{P}n$ and a map for the group $\mathbb{R}^n_{>0}$. To learn the latter map, we construct a network that can translate from the weight space to a diagonal matrix in $\mathbb{R}^n_{>0}$, which can also be represented as a vector in $\mathbb{R}^n$ with all positive entries. We compute statistical features of the weights, with shape $D$, to serve as input to the network. Therefore, the introduced network is basically a MLP $\{W_{\text{scale}}, b_{\text{scale}}\}$ that maps from $D \to n$.

The formula $\sin(W_{\text{scale}}x + b_{\text{scale}}) \cdot \epsilon + 1_n$ naturally relaxes the strict equivariance typically imposed in metanetworks. By introducing a sine function, we allow the transformation to gently oscillate around the identity, creating a smooth, controlled variation. This approach is inspired by Fourier analysis, where sine waves naturally introduce periodic fluctuations, offering flexibility without disrupting the overall structure. The small learnable parameter $\epsilon$ simply scales these oscillations, determining how much relaxation to apply. This design provides a natural way to balance stability and flexibility, enabling the model to remain expressive while avoiding the constraints of strict equivariance.

### B.2 Predicting CNN Generalization

**Dataset.** The Small CNN Zoo provides a $\mathrm{ReLU}$ subset containing 6,050 samples for training and 1,513 for testing. To enlarge the dataset, we apply data augmentation with a factor of 2, generating one additional variant for every original instance. This procedure yields an augmented $\mathrm{ReLU}$ subset with 12,100 training examples and 3,026 test examples.

**Baselines.** We evaluate our model against five established baselines:

- **STATNN** (Unterthiner et al., 2020): extracts statistical features from network weights and biases.

- **Graph-NN** (Kofinas et al., 2024): models network parameters as graphs and applies Graph Neural Networks for processing.

- **NP and HNP** (Zhou et al., 2024a): integrate neuron permutation symmetries into neural functional networks.

- **Monomial-NFN** (Tran et al., 2024a): generalizes the action on weights from permutation matrices to monomial matrices, incorporating scaling and sign-flip symmetries.

**Model Configurations.** Our Monomial-NFN Quasi extends Monomial-NFN, which follows the architecture of Zhou et al. (2024a). The base model consists of three equivariant Monomial-NFN layers with 16, 16, and 5 channels, each followed by a $\mathrm{ReLU}$ activation. In our variant, the final Monomial-NFN layer is replaced with a Monomial-NFN Quasi layer, where a learned scale is applied to the output of the standard Monomial-NFN layer.

The resulting weight-space features are then fed into an invariant Monomial-NFN layer with Monomial-NFN pooling, which contains learnable parameters. This pooling layer normalizes the weights along the hidden dimension and computes averages across rows (first layer), columns (last layer), or both (intermediate layers). The pooled output is flattened and projected to $\mathbb{R}^{200}$, followed by an MLP with two hidden layers and $\mathrm{ReLU}$ activations. Finally, the output is linearly mapped to a scalar and passed through a sigmoid function.

Table 4: Number of parameters of all models for predicting generalization task.

| Model | STATNN | NP | HNP | Monomial-NFN | Monomial-NFN large | Monomial-NFN Quasi |
|---|---|---|---|---|---|---|
| Number of parameters | 1.06M | 2.03M | 2.81M | 0.25M | 0.43 | 0.26M |

Table 5: Hyperparameters for Monomial-NFN and Monomial-NFN an on predicting generalization task.

| | Monomial-NFN dim | MLP dim | Loss | Optimizer | Learning rate | Batch size | Epoch |
|---|---|---|---|---|---|---|---|
| Monomial-NFN | [16,16,5] | 200 | Binary cross-entropy | Adam | 0.001 | 8 | 50 |
| Monomial-NFN large | [20,20,5] | 200 | Binary cross-entropy | Adam | 0.001 | 8 | 50 |
| Monomial-NFN Quasi | [16,16,5 (quasi layer)] | 200 | Binary cross-entropy | Adam | 0.001 | 8 | 50 |

**Training.** We train the model with Binary Cross Entropy (BCE) loss for 50 epochs, applying early stopping based on a validation threshold $\tau$. On an A100 GPU, the full training process requires approximately 35 minutes.

**Other Baselines.** For Graph-NN, we adopt the official implementation provided in (Kofinas et al., 2024) (https://github.com/mkofinas/neural-graphs). The implementations of NP, HNP, and Monomial-NFN follow the setups in (Zhou et al., 2024a) (https://github.com/AllanYangZhou/nfn) and (Tran et al., 2024a). For these models, we use three equivariant layers with channel sizes of 16, 16, and 5. The features extracted are processed through average pooling, and subsequently passed into three MLP layers, each with a hidden size of 200. We provide the number of parameters and hyperparameters in Table 4 and Table 5

### B.3 CLASSIFYING IMPLICIT NEURAL REPRESENTATIONS OF IMAGES

**Dataset.** We employ the original INRs dataset, which includes three subsets: CIFAR-10, MNIST, and Fashion-MNIST. Each image in these datasets is represented using a single SIREN model, as described in Zhou et al. (2024a). Following the setup in Tran et al. (2024a), no data augmentation is applied. The numbers of training, validation, and test samples for each dataset are summarized in Table 6.

**Baselines.** Follow (Zhou et al., 2024a; Tran et al., 2024a), we compare our method against MLP, NP, HNP, and Monomial-NFN. For baselines, we follow the architecture in Zhou et al. (2024a), with a reduced hidden dimension ($512 \rightarrow 256$) to mitigate overfitting. All baseline models and our base model use a hidden dimension of 256. The hyperparameters of Monomial-NFN and the tuned version is given in Table 7

**Model Configurations.** In these experiments, our architecture consists of two Monomial-NFN layers with sine activations, followed by one Monomial-NFN Quasi layer with absolute activation. The hidden dimensions for Monomial-NFN Quasi layers vary by dataset and are listed in Table 8. We also provide the parameters count for all models in Table 9

The subsequent design follows the NP and HNP models of Zhou et al. (2024a). Specifically, we apply a Gaussian Fourier transformation to encode the input with sine and cosine components, expanding from one dimension to 256 dimensions. When using NP as the base layer, the features are further processed by IOSinusoidalEncoding—a positional encoding tailored for NP—with a maximum frequency of 10 and six frequency bands. The encoded features are then passed through three NP or HNP layers with ReLU activations, followed by average pooling. The pooled output is flattened and fed into an MLP with two hidden layers of 1000 units each, also with ReLU activations. Finally, the output is linearly projected to a scalar.

For the MNIST dataset, we insert a Channel Dropout layer (after each HNP ReLU activation) and a Dropout layer (after each MLP ReLU activation), both with a dropout rate of 0.1. Training uses Binary Cross Entropy (BCE) loss for 200,000 steps, which takes approximately 1 hour 50 minutes on an A100 GPU.

Table 6: Dataset size for Classifying INRs task.

|  | Train | Validation | Test |
|---|---|---|---|
| CIFAR-10 | 45000 | 5000 | 10000 |
| MNIST | 45000 | 5000 | 10000 |
| Fashion-MNIST | 45000 | 5000 | 20000 |

Table 7: Hyperparameters of Monomial-NFN and Monomial-NFN tuned (in parentheses) for each dataset in Classify INRs task.

|  | MNIST | Fashion-MNIST | CIFAR-10 |
|---|---|---|---|
| Monomial-NFN hidden dim | 64 (128) | 64 (128) | 16 (64) |
| Base model | HNP | NP | HNP |
| Base model hidden dim | 256 | 256 | 256 |
| MLP hidden neurons | 1000 | 500 | 1000 |
| Dropout value | 0.1 | 0 | 0 |
| Learning rate | 0.000075 | 0.0001 | 0.0001 |
| Batch size | 32 | 32 | 32 |
| Number of training steps | 200000 | 200000 | 200000 |
| Loss function | Binary cross-entropy | Binary cross-entropy | Binary cross-entropy |

Table 8: Hyperparameters of Monomial-NFN Quasi for each dataset in Classify INRs task.

|  | MNIST | Fashion-MNIST | CIFAR-10 |
|---|---|---|---|
| Scale network hidden dim | 32 | 32 | 32 |
| Scale network weight initialization | Xavier | Xavier | Xavier |
| Scale network $\epsilon$ initialization | 0.01 | 0.01 | 0.01 |
| Monomial-NFN hidden dim | 64 | 64 | 16 |
| Base model | HNP | NP | HNP |
| Base model hidden dim | 256 | 256 | 256 |
| MLP hidden neurons | 1000 | 500 | 1000 |
| Dropout value | 0.1 | 0 | 0 |
| Learning rate | 0.000075 | 0.0001 | 0.0001 |
| Batch size | 32 | 32 | 32 |
| Number of training steps | 200000 | 200000 | 200000 |
| Loss function | Binary cross-entropy | Binary cross-entropy | Binary cross-entropy |

Table 9: Number of parameters of all models for classifying INRs task.

|  | CIFAR-10 | MNIST | Fashion-MNIST |
|---|---|---|---|
| MLP | 2M | 2M | 2M |
| NP | 16M | 15M | 15M |
| HNP | 42M | 22M | 22M |
| Monomial-NFN | 16M | 22M | 20M |
| Monomial-NFN tuned | 16.3M | 22.3M | 20.7M |
| Monomial-NFN Quasi (ours) | 16.3M | 22.2M | 20.5M |

## B.4 PREDICTING TRANSFORMERS GENERALIZATION

**Dataset.** We use the Small Transformer Zoo dataset (Tran et al., 2025b), which includes two subsets: MNIST-Transformer and AGNews-Transformer. These datasets are generated by training Transformer models on MNIST image classification and AGNews text classification tasks, while varying key hyperparameters such as training fraction, dropout rate, learning rate, and weight initialization. In total, the zoo contains 62,756 models for MNIST-Transformer and 63,796 models for AGNews-Transformer. Following the experimental setup in (Tran et al., 2025b), we use the checkpoints at epoch 75, corresponding to 15,689 models in MNIST-Transformer and 15,949 models in AGNews-Transformer. The ratio of train, validation and test set is 0.7, 0.15, and 0.15, respectively.

**Baselines.** The baselines are implemented following (Tran et al., 2025b). More specifically:

- **MLP.** For the MLP baseline, each network component is treated independently. The corresponding weights are flattened and passed through dedicated MLPs: a single hidden layer

Table 10: Number of parameters for all models

| Model | MNIST-Transformers | AGNews-Transformers |
|---|---|---|
| Transformer-NFN | 1.812M | 1.804M |
| Transformer-NFN large | 2.857M | 2.857M |
| Transformer-NFN quasi | 1.894M | 1.887M |
| MLP | 0.933M | 0.896M |
| STATNN | 0.203M | 0.168M |

Table 11: Performance measured by Kendall's $\tau$ of all models on AGNews-Transformers dataset. Uncertainties indicate standard error over 5 runs.

| | Quasi dim | No threshold | Accuracy threshold 20% | 40% | 60% | 80% |
|---|---|---|---|---|---|---|
| Transformer-NFN | - | $0.910 \pm 0.001$ | $0.908 \pm 0.001$ | $0.897 \pm 0.001$ | $0.896 \pm 0.001$ | $0.890 \pm 0.001$ |
| Transformer-NFN Quasi | 8 | $0.911 \pm 0.002$ | $0.909 \pm 0.001$ | $0.897 \pm 0.001$ | $0.897 \pm 0.002$ | $0.891 \pm 0.001$ |
| Transformer-NFN Quasi | 16 | $0.913 \pm 0.002$ | $0.911 \pm 0.001$ | $\mathbf{0.901 \pm 0.001}$ | $0.902 \pm 0.002$ | $0.894 \pm 0.001$ |
| Transformer-NFN Quasi | 32 | $\mathbf{0.914 \pm 0.001}$ | $\mathbf{0.913 \pm 0.002}$ | $\mathbf{0.901 \pm 0.001}$ | $\mathbf{0.903 \pm 0.002}$ | $\mathbf{0.896 \pm 0.001}$ |
| Transformer-NFN Quasi | 64 | $0.913 \pm 0.001$ | $0.911 \pm 0.001$ | $0.899 \pm 0.002$ | $\mathbf{0.903 \pm 0.002}$ | $0.894 \pm 0.002$ |

with 50 units is used for both the transformer block and the embedding, while the classifier is modeled by a two-layer MLP with 50 units per layer. The outputs from these modules are concatenated and further processed by a final MLP that produces the prediction.

- **STATNN** (Unterthiner et al., 2020). To adapt STATNN to transformer architectures, we first compute statistical summaries from the weights of the query, key, value, and output projections, together with the weights and biases of the two feedforward layers. These features are concatenated and given as input to a one-layer MLP with 256 hidden units. For the classifier, we retain the original STATNN feature extraction scheme, followed by another MLP with 256 hidden units. The embedding is handled separately with a one-layer MLP of 64 hidden units. The outputs from all three components are merged and passed through a final single-layer MLP to obtain the prediction.

- **XGBoost** (Chen & Guestrin, 2016), **LightGBM** (Ke et al., 2017), and **Random Forest** (Breiman, 2001). For tree-based approaches, we directly flatten all component weights and use them as input features. Across all three models, we set the hyperparameters to a maximum tree depth of 10, a minimum child weight of 50, and at most 256 leaves.

**Model Configurations.** Our approach is built on the Transformer-NFN backbone, with the Transformer-NFN Quasi model consisting of three modules that process the weights of a transformer network. The embedding and classifier components are modeled as MLPs with ReLU activations, each applied independently to their respective inputs. The transformer block is handled separately using an invariant architecture: several equivariant Transformer-NFN Quasi layers with ReLU activations are applied to the two MLP submodules of the block, and their outputs are passed through an invariant polynomial Transformer-NFN layer. The resulting vectors from all components are concatenated and processed by a final MLP with Sigmoid activation to produce the prediction.

In our experiments, the embedding module is implemented as a one-layer MLP with 10 hidden units, while the classifier is a two-layer MLP, each with 10 hidden units. For the transformer block, we use an equivariant Transformer-NFN Quasi layer with 10 hidden channels, followed by an invariant Transformer-NFN layer and an MLP that generates a 10-dimensional output vector. These outputs are concatenated and fed into a classification head to yield the final prediction.

## B.5 ABLATION ON THE MLP NETWORK FOR GROUP ACTION LEARNING

**Experiment Setup.** To assess the robustness of the scale network, we conduct experiments by systematically varying the hidden dimension of the MLPs. Our evaluation is carried out using the Transformer-NFN Quasi architecture on the AGNews-Transformers dataset. In particular, we investigate the capacity of the model to learn the two invertible matrices $M$ and $N$ under different hidden dimensions, specifically 8, 16, 32, and 64.

**Results.** Table 11 reports the performance of Transformer-NFN Quasi under different hidden dimensions of the scale network. Incorporating the quasi-equivariant layer consistently improves performance, with gains increasing as the hidden dimension grows, and reaching the highest score at 32 dimensions. Based on this observation, we select a hidden dimension of 32 for the final model.

### B.6 Experiments on Augmented AGNews-Transformers dataset

**Experiment Setup.** We evaluate the robustness of Transformer-NFN Quasi under strong group actions in weight space, testing whether the quasi layer compromises architectural symmetry. Following the setup in (Tran et al., 2025b), we conduct experiments on the AGNews-Transformers dataset augmented with the group action $\mathcal{G}_{\mathcal{U}}$. Both training and test sets are 2-fold augmented: the original weights are retained, and additional weights are generated by applying permutations and scaling transformations to Transformer modules. The elements of $M$ and $N$ are uniformly sampled from $[-1, 1]$, $[-10, 10]$, and $[-100, 100]$.

**Results.** Table 12 summarizes the results. Transformer-NFN shows stable Kendall's $\tau$ values across all ranges of scaling, with a consistent score of $0.914$. While our model does not gain from increasingly large augmentation scales, its performance remains steady, highlighting the balanced trade-off of Transformer-NFN Quasi: it enhances the expressiveness of the base model while preserving its inherent symmetry.

Table 12: Performance measured by Kendall's $\tau$ of all models on augmented AGNews-Transformers dataset using the group action $\mathcal{G}_{\mathcal{U}}$. Uncertainties indicate standard error over 5 runs.

|  | Original | $[-1, 1]$ | $[-10, 10]$ | $[-100, 100]$ |
|---|---|---|---|---|
| XGBoost | $0.859 \pm 0.001$ | $0.799 \pm 0.003$ | $0.800 \pm 0.001$ | $0.802 \pm 0.003$ |
| LightGBM | $0.835 \pm 0.001$ | $0.785 \pm 0.003$ | $0.784 \pm 0.003$ | $0.786 \pm 0.004$ |
| Random Forest | $0.774 \pm 0.003$ | $0.714 \pm 0.001$ | $0.715 \pm 0.002$ | $0.716 \pm 0.002$ |
| MLP | $0.879 \pm 0.006$ | $0.830 \pm 0.002$ | $0.833 \pm 0.002$ | $0.833 \pm 0.005$ |
| STATNN | $0.841 \pm 0.002$ | $0.793 \pm 0.003$ | $0.791 \pm 0.003$ | $0.771 \pm 0.013$ |
| Transformer-NFN | $0.910 \pm 0.001$ | $0.912 \pm 0.001$ | $0.912 \pm 0.002$ | $0.913 \pm 0.001$ |
| Transformer-NFN Quasi | $\mathbf{0.914 \pm 0.001}$ | $\mathbf{0.914 \pm 0.002}$ | $\mathbf{0.914 \pm 0.002}$ | $\mathbf{0.914 \pm 0.002}$ |

### B.7 Analysis on the learned scaling

**Experiment Setup.** To analyze the sensitivity of $\epsilon$ and the learned scaling $(\sin(\gamma(\theta)) \cdot \epsilon + 1_n)$ in MLP/CNN case, we conduct experiments on the task of Predicting CNN Generalization. We introduce learned scale layers (with corresponding $\epsilon$) on top of Monomial-NFN equivariant layers and explore two cases: one with a fixed $\epsilon$ and one with a learnable $\epsilon$. The results are presented in Table 13. The table reports the following values:

- Initial $\epsilon$: The initial value of $\epsilon$

- Learnable/Fixed: How $\epsilon$ changes during training

- Final $\epsilon$: The final value of the first layer $\epsilon$ after training

- Learned scale: The first layer learned scaling $(\sin(W_{\text{scale}}x + b_{\text{scale}}) \cdot \epsilon + 1_n)$

- Kendall's $\tau$: Evaluation metric (Higher is better)

**Results.** When $\epsilon$ is fixed at a small value, the learned scaling slightly deviates from the identity, resulting in marginal performance gains. However, when $\epsilon$ is learnable, the learned scaling can deviate further from the identity, leading to a broader scaling range that enhances performance. Additionally, when $\epsilon$ is learnable, the final $\epsilon$ values tend to converge, regardless of their initial values. For larger values of $\epsilon$, instability may arise due to significant deviations in the early steps. Consequently, in our study, we choose a learnable $\epsilon$ with an initial value of 0.01 to ensure training stability.

Table 13: Analysis on the learned scaling with varying $\epsilon$

| Initial $\epsilon$ | Learnable/Fixed | Final $\epsilon$ | Learned scale | Kendall's $\tau$ |
|---|---|---|---|---|
| 0 (Baseline) | - | - | - | 0.922 |
| 0.001 | Fixed | 0.001 | $1.000 \pm 0.001$ | 0.922 |
| 0.01 | Fixed | 0.01 | $0.999 \pm 0.007$ | 0.923 |
| 0.1 | Fixed | 0.1 | $1.003 \pm 0.079$ | 0.923 |
| 0.2 | Fixed | 0.2 | $0.933 \pm 0.135$ | 0.923 |
| 0.4 | Fixed | 0.4 | $0.972 \pm 0.378$ | 0.924 |
| 0.001 | Learnable | 0.966 | $0.796 \pm 0.668$ | 0.925 |
| 0.01 | Learnable | 0.970 | $0.941 \pm 0.710$ | 0.926 |
| 0.1 | Learnable | 0.971 | $0.918 \pm 0.758$ | 0.926 |
| 0.2 | Learnable | 0.946 | $0.743 \pm 0.626$ | 0.925 |
| 0.4 | Learnable | 0.988 | $0.762 \pm 0.624$ | 0.925 |

## B.8 SENSITIVITY OF $\epsilon$

**Experiment Setup.** We present an ablation study on the sensitivity of $\epsilon$ by conducting the "Predict Transformer Generalization" experiment on the MNIST-Transformer dataset. Specifically, we analyze two cases: when $\epsilon$ is learnable versus fixed, and report the performance alongside statistics of learned noises $(\sin(\gamma(\theta)) \cdot \epsilon)$ for the matrices $M$ and $N$, which represent the two $GL$ group actions applied to the weights of the MHA layer. For each noise, we compute the mean and standard deviation of the diagonal and off-diagonal elements. The results are summarized in Table 14. The table reports the following values:

- Initial $\epsilon$: The initial value of $\epsilon$ for both $M$ and $N$.

- Learnable/Fixed: Whether $\epsilon$ is changed during training.

- Final $\epsilon_M$, $\epsilon_N$: The final value of $\epsilon$ after training.

- Diagonal $M$, $N$: Mean and standard deviation of the learned noises $\sin(\gamma(\theta)) \cdot \epsilon$ for $M$ and $N$, computed for diagonal elements.

- Off-diagonal $M$, $N$: Mean and standard deviation of the learned noises $\sin(\gamma(\theta)) \cdot \epsilon$ for $M$ and $N$, computed for off-diagonal elements.

- Kendall's $\tau$: Evaluation metric (higher is better).

Table 14: Ablation study on $\epsilon$ with metrics for M and N.

| Initial $\epsilon$ | Type | Final $\epsilon_M$ | Final $\epsilon_N$ | Diagonal M | Off-diagonal M | Diagonal N | Off-diagonal N | Kendall's $\tau$ |
|---|---|---|---|---|---|---|---|---|
| Baseline | - | - | - | - | - | - | - | 0.905 |
| 0.001 | Fixed | 0.001 | 0.001 | $0.000 \pm 0.000$ | $0.062 \pm 0.242$ | $0.000 \pm 0.000$ | $0.063 \pm 0.242$ | 0.909 |
| 0.01 | Fixed | 0.01 | 0.01 | $0.001 \pm 0.002$ | $0.062 \pm 0.242$ | $-0.001 \pm 0.005$ | $0.062 \pm 0.242$ | 0.909 |
| 0.1 | Fixed | 0.1 | 0.1 | $0.005 \pm 0.020$ | $0.062 \pm 0.245$ | $-0.038 \pm 0.065$ | $0.061 \pm 0.244$ | 0.907 |
| 0.2 | Fixed | 0.2 | 0.2 | $0.010 \pm 0.040$ | $0.061 \pm 0.249$ | $-0.002 \pm 0.144$ | $0.059 \pm 0.285$ | 0.910 |
| 0.4 | Fixed | 0.4 | 0.4 | $0.021 \pm 0.080$ | $0.060 \pm 0.265$ | $-0.055 \pm 0.221$ | $0.073 \pm 0.323$ | 0.909 |
| 0.001 | Learnable | 0.001 | -0.187 | $0.000 \pm 0.000$ | $0.063 \pm 0.242$ | $-0.068 \pm 0.117$ | $0.059 \pm 0.260$ | 0.908 |
| 0.01 | Learnable | 0.010 | 0.182 | $0.001 \pm 0.002$ | $0.062 \pm 0.241$ | $-0.056 \pm 0.084$ | $0.061 \pm 0.244$ | 0.911 |
| 0.1 | Learnable | 0.099 | 0.234 | $0.005 \pm 0.020$ | $0.062 \pm 0.245$ | $-0.085 \pm 0.114$ | $0.060 \pm 0.250$ | 0.910 |
| 0.2 | Learnable | 0.200 | 0.290 | $0.010 \pm 0.040$ | $0.061 \pm 0.249$ | $-0.011 \pm 0.218$ | $0.062 \pm 0.316$ | 0.909 |
| 0.4 | Learnable | 0.400 | 0.298 | $0.021 \pm 0.079$ | $0.060 \pm 0.265$ | $-0.011 \pm 0.179$ | $0.065 \pm 0.298$ | 0.909 |

**Results.** When the $GL$ group is learned through our quasi-equivariant layers, the off-diagonal elements provide additional flexibility, leading to a noticeable improvement in performance even with a fixed $\epsilon$. With a learnable $\epsilon$, the range of values for $N$ expands and become more stable, resulting in better predictive performance. Overall, this study suggests that increasing the relaxation through $\epsilon$ can enhance performance. For our experiments, we select a learnable $\epsilon$ with an initial value of 0.01 to ensure the training stability.

## B.9 Weight space style editing

**Experiment Setup.**    In this experiment, we focus on modifying the image content encoded in each SIREN model (Sitzmann et al., 2020) by adjusting its weights. Specifically, we leverage pretrained models from (Zhou et al., 2024a), which encode images from the CIFAR-10 and MNIST datasets. The experimental setup follows the approach described in (Zhou et al., 2024a; Tran et al., 2024a). Our main goal is to explore two tasks that involve altering the embedded information within the SIREN model: first, enhancing the contrast of CIFAR-10 images, and second, applying dilation to MNIST images.

To evaluate the effectiveness of these modifications, we compute the mean squared error (MSE) loss between the images encoded by the modified SIREN network and the corresponding ground truth images, which have undergone contrast enhancement for CIFAR-10 and dilation for MNIST. The baseline for comparison is the Monomial-NFN model (Tran et al., 2024a), which is already a highly optimized version. In this case, further increasing the parameters does not lead to a noticeable improvement in performance, serving as a useful benchmark for our experiments. The results from these tasks are summarized in Table 15.

Table 15: Weight space style editing (INRs editing)

| Model | Contrast (CIFAR-10) | Dilate (MNIST) |
|---|---|---|
| MLP | $0.031 \pm 0.001$ | $0.306 \pm 0.001$ |
| HNP (Zhou et al., 2024a) | $0.021 \pm 0.001$ | $0.071 \pm 0.001$ |
| NP (Zhou et al., 2024a) | $0.020 \pm 0.002$ | $0.068 \pm 0.002$ |
| Monomial-NFN (Tran et al., 2024a) | $0.020 \pm 0.001$ | $0.069 \pm 0.002$ |
| Monomial-NFN Quasi (1% params ++) | $\mathbf{0.019 \pm 0.001}$ | $\mathbf{0.066 \pm 0.001}$ |

**Results.**    In this task, Monomial-NFN model initially underperforms compared to the NP model. However, by incorporating our newly introduced Quasi layer, which adds only 1% more parameters, we are able to significantly improve the performance of the Monomial-NFN model. This enhancement allows it to surpass the performance of the NP baseline in both tasks, demonstrating the effectiveness of our approach with minimal increase in model complexity.

