# OpenReview forum: "Quasi-Equivariant Metanetworks"
_ICLR.cc/2026/Conference — ICLR 2026 Poster_

### Official Review · Reviewer_J9ur · 2025-10-24

**Soundness:** 3
**Presentation:** 4
**Contribution:** 3
**Rating:** 8
**Confidence:** 4

**Summary:**

This paper introduces the concept of quasi-equivariance specifically for metanetworks, a novel framework designed to overcome the limitations of strict equivariance in metanetworks. Metanetworks are models that process the parameters of other neural networks, and a key challenge is functional equivalence—where different parameter sets can yield the same input-output function. While strictly equivariant metanetworks respect the symmetries underlying functional equivalence (e.g., neuron permutations), they can be overly restrictive and lack expressivity.

This work proposes quasi-equivariance as a principled relaxation. A map $F$ is quasi-equivariant with respect to a symmetry group $G$ if for any transformation $g \in G$ applied to its input $\theta$, its output is transformed by some other group element $g' \in G$, i.e., $F(g\theta) = g'F(\theta)$. This ensures the output remains in the same functional equivalence class, thus preserving functional identity, while allowing for greater flexibility than the strict condition $F(g\theta) = gF(\theta)$.

The paper provides a theoretical foundation for this concept, connecting it to the notion of a maximal symmetry group, and proposes a constructive method for building quasi-equivariant layers. This is achieved by augmenting a standard equivariant map $\beta(\theta)$ with a learned, parameter-dependent group action $\alpha(\theta)$, yielding a layer of the form $F(\theta) = \alpha(\theta)\beta(\theta)$. The framework's effectiveness is demonstrated by applying it to feedforward, convolutional, and transformer networks. Empirical results across three distinct metanetwork tasks—predicting model generalization, classifying implicit neural representations, and predicting transformer performance—show that quasi-equivariant layers significantly improve the performance of state-of-the-art metanetworks with only a minimal increase in parameter count.

**Strengths:**

Originality: This paper introduces the novel concept of "quasi-equivariance" for metanetworks, which is a significant and original extension to the existing paradigm of strict equivariance. This relaxation addresses a critical limitation of previous approaches, offering a new direction for designing more flexible and expressive models.

Quality: This paper provides a strong theoretical foundation for quasi-equivariance, connecting it to functional equivalence and maximal symmetry groups. The empirical evaluations are thorough, comparing the proposed method against several established baselines across diverse tasks, datasets, and architectures that the metanetwork processes, consistently demonstrating its effectiveness.

Clarity: This paper is overall well-written and well-structured, with clear explanations of complex concepts such as functional equivalence, symmetry groups, and the mathematical definitions of equivariance, supported by figures and tables of results.

Significance: The introduction of quasi-equivariance has significant implications for the design of metanetworks, enabling better trade-offs between symmetry preservation and representational flexibility. The empirical results demonstrate substantial performance improvements with minimal additional cost.

**Weaknesses:**

Scope Limitations:
* Your proposed methods framework are only demonstrated on MLP metanetworks that operate on the parameters of MLPs, CNNs, and Transformers, whose symmetry groups have a relatively simple direct-product structure. You briefly mention that extending this framework to more complex architectures like graph-based metanetworks are only briefly mentioned, but expanding upon what the challenges are and how they might be tackled in a few sentences could provide better context for the audience.
* You provide clear evidence that the proposed quasi-equivariant framework is useful for metanetworks, but are there other fields of machine learning that could benefit from it? Adding a sentence with some ideas would help broaden the audience of this paper.

The implementation for MLPs and CNNs in Section 4.2 is currently only provided for MLPs, but the experiments focus on CNNs. Can you add a few sentences on how the provided MLP implementation translates to the CNN case?

Although $\text{GL}(d_h)$ is defined in the notation table, it would be worth also defining around line 344.

The text is table 3 is quite small and difficult to read. Please make it at least the same size as the text in the other tables.

Formatting: On line 066, use \citet instead of \citep to get "...class of MLPs, Fefferman & Markel (1993) established..."

**Questions:**

The weaknesses above are important to address in the main paper, even given the space constraints. The following questions can be addressed in an appendix and referred to in the main paper.

Have you performed any analysis of the learned group elements from the map $\alpha(\theta)$ in the experiments? For instance, how far from the identity do the learned scaling factors or matrices tend to be, especially in the case of the experiments presented in section 5.1?

The construction of the map to $GL(n)$ via $\sin(\gamma(\theta)) \cdot \epsilon+ I_n$ is an interesting and practical choice. How sensitive is the model's performance to the hyperparameter $\epsilon$?

---

> ### Author Response · Authors · 2025-11-19
>
> We thank the Reviewer for the response and address the concerns below. The Reviewer may also refer to our General Response for a summary of the revisions.
>
> ---
>
> **W1. Your proposed methods framework are only demonstrated on MLP metanetworks that operate on the parameters of MLPs, CNNs, and Transformers, whose symmetry groups have a relatively simple direct-product structure. You briefly mention that extending this framework to more complex architectures like graph-based metanetworks are only briefly mentioned, but expanding upon what the challenges are and how they might be tackled in a few sentences could provide better context for the audience.**
>
> **Answer W1.** We thank the Reviewer for the comment. As discussed in the manuscript, quasi-equivariance is, in principle, a theoretically optimal relaxation of strict equivariance in the context of metanetworks. However, applying quasi-equivariance to graph-based metanetworks is nontrivial. To the best of our knowledge, existing graph-based metanetworks primarily handle permutation symmetry, and only a few works consider broader symmetries such as scaling~[1]. We were also unable to identify a practical way to incorporate full quasi-equivariance into current graph-based formulations. For this reason, we keep that sentence as pointing toward a potential future direction for this line of research.
>
> *References*
>
> [1] Ioannis Kalogeropoulos et al., Scale Equivariant Graph Metanetworks
>
> **W2. You provide clear evidence that the proposed quasi-equivariant framework is useful for metanetworks, but are there other fields of machine learning that could benefit from it? Adding a sentence with some ideas would help broaden the audience of this paper.**
>
> **Answer W2.** Thank you for your suggestion. The ability to apply quasi-equivariance in other fields is indeed crucial, as it allows for more flexible and robust modeling, particularly in domains where strict symmetry constraints may not always hold. In areas like computational chemistry, physics, and materials science, many models are designed to respect physical symmetries--such as rotations, translations, and reflections--yet in practical scenarios, these symmetries often only approximately hold, or need to be relaxed for more realistic modeling. A quasi-equivariant approach can provide the flexibility to preserve functional symmetry while allowing for deviations when necessary, improving model expressivity and robustness. We have incorporated this discussion into the Conclusion section of the manuscript.
>
> **W3. The implementation for MLPs and CNNs in Section 4.2 is currently only provided for MLPs, but the experiments focus on CNNs. Can you add a few sentences on how the provided MLP implementation translates to the CNN case?**
>
> **Answer W3.** In the MLP setup, the quasi-equivariant layer learns a positive scaling vector for each layer by applying a group action to the output dimensions of the weights and biases. The group action is parameterized as a scaling factor learned per layer.
>
> For CNNs, the approach follows the same principle but is adapted for convolutional filters. Each bias vector $b^{(i)}$ retains the same dimensions as in MLPs, while the convolution filter $W^{(i)} \in \mathbb{R}^{n_i \times n_{i-1} \times w}$ (for 1D convolution) or $W^{(i)} \in \mathbb{R}^{n_i \times n_{i-1} \times c \times w}$ (for 2D convolution) contains additional spatial dimensions. To handle this, we apply the group action to both the filter's channel dimensions and spatial dimensions. Specifically, the filter $W^{(i)}$ is treated as having dimensions $n_i \times n_{i-1} \times (c w)$, where $c$ represents the number of input channels (or more generally, any extra spatial dimensions like spatial channels). This allows us to apply the quasi-equivariant layer to the filter in a manner similar to the MLP case, where we perform the scaling operation across the output channels and input channels.
>
>
> **W4. Although $GL(d_h)$ is defined in the notation table, it would be worth also defining around line 344.**
>
> **Answer W4.** Thank you for your suggestion. We have updated the manuscript accordingly.
>
> **W5. The text is table 3 is quite small and difficult to read. Please make it at least the same size as the text in the other tables.**
>
> **Answer W5.** Thank you for pointing out this. We have updated the manuscript accordingly.
>
> **W6. Formatting: On line 066, use \citet instead of \citep to get "...class of MLPs, Fefferman & Markel (1993) established..."**
>
> **Answer W6.** Thank you for pointing out this. We have updated the manuscript accordingly.

---

> > ### Author Response · Authors · 2025-11-19
> >
> > **Q1. Have you performed any analysis of the learned group elements from the map $\alpha(\theta)$ in the experiments? For instance, how far from the identity do the learned scaling factors or matrices tend to be, especially in the case of the experiments presented in section 5.1?**
> >
> > **Answer Q1.** Following your suggestion, we provide an analysis of the learned scaling ($\operatorname{sin}(W_{\text{scale}} x + b_{\text{scale}}) \cdot \epsilon + 1_n$). Specifically, we conduct experiments on the task of Predicting CNN Generalization, where we introduce learned scale layers (with corresponding $\epsilon$) on top of Monomial-NFN equivariant layers. We explore two cases: one with a fixed $\epsilon$ and one with a learnable $\epsilon$. The results are presented in Table 1 (Table 13 of Appendix B.7 in our revised version). The table reports the following values:
> >
> > - Initial $\epsilon$: The initial value of $\epsilon$
> > - Learnable/Fixed: How $\epsilon$ changes during training
> > - Final $\epsilon$: The final value of the first layer $\epsilon$ after training
> > - Learned scale: The first layer learned scaling ($\operatorname{sin}(W_{\text{scale}} x + b_{\text{scale}}) \cdot \epsilon + 1_n$)
> > - Kendall's $\tau$: Evaluation metric (Higher is better)
> >
> > **Table 1: Analysis on the learned scaling with varying $\epsilon$**
> > | Initial $\epsilon$ | Learnable/Fixed | Final $\epsilon$ | Learned scale | Kendall's $\tau$ |
> > |------------|-----------------|--------------------|--------------------|--------------------|
> > | 0 (Baseline)      | -          | -             | -    | 0.922            |
> > | 0.001      | Fixed           | 0.001              | $1.000\pm0.001$    | 0.922            |
> > | 0.01       | Fixed           | 0.01              | $0.999\pm0.007$    | 0.923            |
> > | 0.1        | Fixed           | 0.1                | $1.003\pm0.079$    | 0.923            |
> > | 0.2        | Fixed           | 0.2                | $0.933\pm0.135$    | 0.923            |
> > | 0.4        | Fixed           | 0.4                | $0.972\pm0.378$   | 0.924            |
> > | 0.001      | Learnable       | 0.966              | $0.796\pm0.668$     | 0.925            |
> > | 0.01       | Learnable       | 0.970              | $0.941\pm0.710$     | 0.926            |
> > | 0.1        | Learnable       | 0.971              | $0.918\pm0.758$     | 0.926            |
> > | 0.2        | Learnable       | 0.946              | $0.743\pm0.626$   | 0.925            |
> > | 0.4        | Learnable       | 0.988              | $0.762\pm0.624$    | 0.925            |
> >
> >
> > When $\epsilon$ is fixed at a small value, the learned scaling slightly deviates from the identity, resulting in marginal performance gains. However, when $\epsilon$ is learnable, the learned scaling can deviate further from the identity, leading to a broader scaling range that enhances performance. Additionally, when $\epsilon$ is learnable, the final $\epsilon$ values tend to converge, regardless of their initial values. For larger values of $\epsilon$, instability may arise due to significant deviations in the early steps. Consequently, in our study, we choose a learnable $\epsilon$ with an initial value of 0.01 to ensure training stability.

---

> > > ### Author Response · Authors · 2025-11-19
> > >
> > > **Q2. The construction of the map to $GL(n)$ via $\operatorname{sin}(\gamma(\theta))\cdot \epsilon + I_n$ is an interesting and practical choice. How sensitive is the model's performance to the hyperparameter $\epsilon$?**
> > >
> > > **Answer Q2.** We present an ablation study on the sensitivity of $\epsilon$ by conducting the "Predict Transformer Generalization" experiment on the MNIST-Transformer dataset. Specifically, we analyze two cases: when $\epsilon$ is learnable versus fixed, and report the performance alongside statistics of learned noises ($\operatorname{sin}(\gamma(\theta)) \cdot \epsilon$) for the matrices $M$ and $N$, which represent the two $GL$ group actions applied to the weights of the MHA layer. For each noise, we compute the mean and standard deviation of the diagonal and off-diagonal elements. The results are summarized in Table 2 (Table 14 of Appendix B.8 in our revised version). The table reports the following values:
> > >
> > > - Initial $\epsilon$: The initial value of $\epsilon$ for both $M$ and $N$.
> > > - Learnable/Fixed: Whether $\epsilon$ is changed during training.
> > > - Final $\epsilon_M$, $\epsilon_N$: The final value of $\epsilon$ after training.
> > > - Diagonal $M$, $N$: Mean and standard deviation of the learned noises $\operatorname{sin}(\gamma(\theta)) \cdot \epsilon$ for $M$ and $N$, computed for diagonal elements.
> > > - Off-diagonal $M$, $N$: Mean and standard deviation of the learned noises $\operatorname{sin}(\gamma(\theta)) \cdot \epsilon$ for $M$ and $N$, computed for off-diagonal elements.
> > > - Kendall's $\tau$: Evaluation metric (higher is better).
> > >
> > > **Table 2: Sensitivity of $\epsilon$**
> > >
> > > |Initial $\epsilon$|Learnable/Fixed|Final $\epsilon_M$|Final $\epsilon_N$|Diagonal $M$|Off-diagonal $M$|Diagonal $N$|Off-diagonal $N$|Kendall's $\tau$|
> > > |-|-|-|-|-|-|-|-|-|
> > > |0 (Baseline)|-|-|-|-|-|-|-|0.905|
> > > |0.001|Fixed|0.001|0.001|$0.000\pm0.000$|$0.062\pm0.242$|$0.000\pm0.000$|$0.063\pm0.242$|0.909|
> > > |0.01|Fixed|0.01|0.01|$0.001\pm0.002$|$0.062\pm0.242$|$-0.001\pm0.005$|$0.062\pm0.242$|0.909|
> > > |0.1|Fixed|0.1|0.1|$0.005\pm0.020$|$0.062\pm0.245$|$-0.038\pm0.065$|$0.061\pm0.244$|0.907|
> > > |0.2|Fixed|0.2|0.2|$0.010\pm0.040$|$0.061\pm0.249$|$-0.002\pm0.144$|$0.059\pm0.285$|0.910|
> > > |0.4|Fixed|0.4|0.4|$0.021\pm0.080$|$0.060\pm0.265$|$-0.055\pm0.221$|$0.073\pm0.323$|0.909|
> > > |0.001|Learnable|0.001|-0.187|$0.000\pm0.000$|$0.063\pm0.242$|$-0.068\pm0.117$|$0.059\pm0.260$|0.908|
> > > |0.01|Learnable|0.010|0.182|$0.001\pm0.002$|$0.062\pm0.241$|$-0.056\pm0.084$|$0.061\pm0.244$|0.911|
> > > |0.1|Learnable|0.099|0.234|$0.005\pm0.020$|$0.062\pm0.245$|$-0.085\pm0.114$|$0.060\pm0.250$|0.910|
> > > |0.2|Learnable|0.200|0.290|$0.010\pm0.040$|$0.061\pm0.249$|$-0.011\pm0.218$|$0.062\pm0.316$|0.909|
> > > |0.4|Learnable|0.400|0.298|$0.021\pm0.079$|$0.060\pm0.265$|$-0.011\pm0.179$|$0.065\pm0.298$|0.909|
> > >
> > > When the $GL$ group is learned through our quasi-equivariant layers, the off-diagonal elements provide additional flexibility, leading to a noticeable improvement in performance even with a fixed $\epsilon$. With a learnable $\epsilon$, the range of values for $N$ expands and becomes more stable, resulting in better predictive performance. Overall, this study suggests that increasing the relaxation through $\epsilon$ can enhance performance. For our experiments, we select a learnable $\epsilon$ with an initial value of 0.01 to ensure training stability.
> > >
> > > ---
> > >
> > > We thank the Reviewer for the constructive feedback and thoughtful suggestions. We remain open to further discussion during the next stage of discussion.

---

> > ### Comment · Reviewer_J9ur · 2025-11-24
> > **Response to Authors' Comment**
> >
> > I thank the authors for their careful revisions and responses to each reviewer. Regarding my own review and it's respective responses, I appreciate the detailed new experimental results.
> >
> > W2: I assume that you are talking about networks that are not meta-networks? I think this distinction is a good point to make. You cite a small sampling of methods for approximate equivariance in non-meta-networks: are any of these methods worth applying to the meta-network vase to compare results with?
> >
> > W3: I appreciate your explanation here, but as your audience may have the same question, can you also incorporate this into the paper?

---

> ### Author Response · Authors · 2025-11-24
>
> We thank the Reviewer for the further concern and happy that Reviewer appreciate our additional experiments. We now address the Reviewer’s further concern.
>
> ---
>
> **W2. I assume that you are talking about networks that are not meta-networks? I think this distinction is a good point to make. You cite a small sampling of methods for approximate equivariance in non-meta-networks: are any of these methods worth applying to the meta-network vase to compare results with?**
>
> **Answer W2.** Indeed, outside the context of metanetworks, several variants of equivariance have been introduced in the literature. In our paper, we summarize two representative forms of such equivariance in Remark 3.4. However, these variants are not well suited for metanetworks, for the following reasons.
>
> - $\varphi(gx) = g' \varphi(x)$ for some $g' \in G_x$.
> This variant is already subsumed by our definition. The additional requirement $g' \in G_x$ (instead of $g' \in G$) does not play any meaningful role and therefore does not introduce a distinct or useful relaxation in our setting.
> - $\varphi(gx) \approx g\varphi(x)$.
> At first sight, this approximation could be applied to metanetworks. However, the approximation concerns $F$ itself. After composing with $\rho$, the resulting map $\rho \circ F$ generally cannot achieve any reasonable form of "functional-preserving approximation", because this depends heavily on how the model $f$ is parameterized. For instance, in an MLP $f(\cdot;\theta)$, having $\theta \approx \theta'$ does not imply that $f(\cdot;\theta)$ approximates $f(\cdot;\theta')$ in any meaningful sense, unless one further analyzes the continuity or stability properties of the parametrization. A secondary issue is that approximate equivariance in the literature typically concerns simple groups such as $S_n$, $\mathrm{SO}(n)$, $\mathrm{O}(n)$, or $\mathrm{GL}(n)$, whereas the symmetry groups of neural architectures are often large products of groups. This makes such relaxed notions either inefficient or inapplicable in the metanetwork setting.
>
> For these reasons, we do not adopt or implement other forms of relaxed equivariance in our metanetwork framework.
>
>
> **W3. I appreciate your explanation here, but as your audience may have the same question, can you also incorporate this into the paper?**
>
> **Answer W3.** We sincerely appreciate your suggestion to clarify this point for a broader audience. We have now incorporated this discussion into Remark 4.2 (lines 324–332) in the main text of the revised manuscript.
>
> ---
>
> We thank the Reviewer for engaging in further discussion of our work and sincerely appreciate the effort invested in helping us improve the manuscript. We remain open to continuing the discussion if the Reviewer has any further concerns.

---

### Official Review · Reviewer_9pd6 · 2025-10-31

**Soundness:** 2
**Presentation:** 2
**Contribution:** 2
**Rating:** 6
**Confidence:** 4

**Summary:**

This work posits the question whether strict equivariance is necessary for metanetworks, and introduces the notion of quasi-equivariance, allowing metanetworks to move beyond strict equivariance. The authors demonstrate the effectiveness of their method in a suite of experiments that includes INR classification, and predicting CNN/Transformer generalization.

**Strengths:**

The manuscript is well-written, and the math formalism is mostly clear.

The research question posed, i.e. whether strict equivariance is necessary for metanetworks is a very topical and significant question.

The proposed method is simple and has a small computational overhead.

**Weaknesses:**

The method details (namely the design of the quasi-equivariant layer) are not described in detail and are, thus, unclear.

A large part of the manuscript is used to re-iterate existing group theory and the math behind parameter space, which obfuscates the novelty of the proposed work.

The datasets used are somewhat saturated, and the performance gains are often marginal.

There are no ablation studies to examine the significance of various hyperparameters.

**Questions:**

1. The experiments are only focusing on invariant tasks. Why don't the authors perform experiments on equivariant tasks (e.g. INR editing)?

2. The performance reported for baselines is often different from the original works (e.g. the works of Zhou et al., Kofinas et al.). In the case of INR classification, specifically, the performance gap is quite large. Why is that?

3. The performance gains of the proposed method compared to the baselines are often marginal. Are there any scenarios/experiments where the proposed method can be expected to boost the baseline greatly? If so, can that be shown through an experiment?

---

> ### Author Response · Authors · 2025-11-19
>
> We thank the Reviewer for the response and address the concerns below. For clarity and coherence, we found it appropriate to merge certain related weaknesses and questions and provide unified answers. The Reviewer may also refer to our General Response for a summary of the revisions.
>
> ---
>
> **W1. The method details (namely the design of the quasi-equivariant layer) are not described in detail and are, thus, unclear.**
>
> **Answer W1.**  We thank the Reviewer for this comment. We provide a more detailed explanation below.
>
> In the MLP/CNN case, the map $\alpha$ consists of two parts: a constant map for the group $\mathcal{P}n$ and a map for the group $\mathbb{R}\_{>0}^n$. To learn the latter map, we construct a network that can translate from the weight space to a diagonal matrix in $\mathbb{R}\_{>0}^n$, which can also be represented as a vector in $\mathbb{R}^{n}$ with all positive entries. We compute statistical features of the weights, with shape $D$, to serve as input to the network. Therefore, the introduced network is basically a MLP $\{W_{\text{scale}}, b\_{\text{scale}}\}$ that maps from $D\to n$.
>
> The formula $\operatorname{sin}(W_{\text{scale}} x + b_{\text{scale}}) \cdot \epsilon + 1_n$ naturally relaxes the strict equivariance typically imposed in metanetworks. By introducing a sine function, we allow the transformation to gently oscillate around the identity, creating a smooth, controlled variation. This approach is inspired by Fourier analysis, where sine waves naturally introduce periodic fluctuations, offering flexibility without disrupting the overall structure. The small learnable parameter $\epsilon$ simply scales these oscillations, determining how much relaxation to apply. This design provides a natural way to balance stability and flexibility, enabling the model to remain expressive while avoiding the constraints of strict equivariance.
>
> We have incoporated this explanation to the end of Appendix B.1 (line 1144-1156) in our revised manuscript.
>
>
> **W2. A large part of the manuscript is used to re-iterate existing group theory and the math behind parameter space, which obfuscates the novelty of the proposed work.**
>
> **Answer W2.** The content in Sections 2 and 3, which involves some background in group theory, is essential for motivating our main contributions. These sections introduce the notion of the maximal symmetry group and the motivation for quasi-equivariance. Without this foundation, the rationale behind using equivariance in metanetwork design would be unclear, as would the need to consider the maximal symmetry group or to introduce quasi-equivariance as an appropriate and principled relaxation of full equivariance. To the best of our knowledge, these conceptual points are novel and form a core part of our contribution.
>
> We also emphasize that the examples provided in these sections are intentionally included to guide readers who may not have a background in group theory. While they may appear redundant to some readers, they play an important role in ensuring accessibility and conveying the intuition behind the concepts. For this reason, we believe the space dedicated to these explanations does not overcomplicate the paper, but rather supports clarity and strengthens the motivation for our framework.
>
> We hope this explanation clarifies why these sections are necessary components of the main manuscript.
>
>
> **W3 + Q3. The datasets used are somewhat saturated, and the performance gains are often marginal. Are there any scenarios/experiments where the proposed method can be expected to boost the baseline greatly? If so, can that be shown through an experiment?**
>
> **Answer W3+Q3.** We followed the experimental setups from recent metanetwork research [1,2,3], utilizing popular datasets in metanetworks learning: Small CNN Zoo (2020) [4], SIRENs dataset (2023) [1], Small Transformer Zoo (2025) [3].
>
> Our results demonstrate the efficiency and effectiveness of the proposed quasi-equivariant layer. Despite adding only 3-5% more parameters, this plug-in module can enhance the performance of existing strict-equivariant metanetworks in a non-trivial way. For instance, in the task of Predicting CNN Generalization, scaling up the existing Monomial-NFN architecture with 68% more parameters results in only a 0.001 improvement in Kendall's $\tau$ on the original ReLU subset. In contrast, with just an additional 4% in parameters, our quasi-equivariant layer improves performance by 0.004, matching the baseline performance of HNP.

---

> > ### Author Response · Authors · 2025-11-19
> >
> > **W4. There are no ablation studies to examine the significance of various hyperparameters.**
> >
> > **Answer W4.** Thank you for your comment. Below, we present an ablation study from our original manuscript and two newly conducted ablation studies to examine the significance of hyperparameters.
> >
> > *Ablation on the hidden dimension of MLP*
> >
> > In Appendix B.5 of our manuscript, we provide an ablation study on the hidden dimension of the MLP network used to learn the group action. Our evaluation is carried out using the Transformer-NFN Quasi architecture on the AGNews-Transformers dataset. In particular, we investigate the model's capacity to learn the two invertible matrices (M) and (N) under different hidden dimensions, specifically 8, 16, 32, and 64. Our results (presented in Table 11 in the manuscript) show that incorporating the quasi-equivariant layer consistently improves performance, with gains increasing as the hidden dimension grows, reaching the highest score at 32 dimensions.
> >
> >
> > *Ablation on the $\epsilon$ in MLP/CNN case*
> >
> > Following your suggestion, we provide an analysis of the learned scaling ($\operatorname{sin}(W_{\text{scale}} x + b_{\text{scale}}) \cdot \epsilon + 1_n$). Specifically, we conduct experiments on the task of Predicting CNN Generalization, where we introduce learned scale layers (with corresponding $\epsilon$) on top of Monomial-NFN equivariant layers. We explore two cases: one with a fixed $\epsilon$ and one with a learnable $\epsilon$. The results are presented in Table 1 (Table 13 of Appendix B.7 in our revised version). The table reports the following values:
> >
> > - Initial $\epsilon$: The initial value of $\epsilon$
> > - Learnable/Fixed: How $\epsilon$ changes during training
> > - Final $\epsilon$: The final value of the first layer $\epsilon$ after training
> > - Learned scale: The first layer learned scaling ($\operatorname{sin}(W_{\text{scale}} x + b_{\text{scale}}) \cdot \epsilon + 1_n$)
> > - Kendall's $\tau$: Evaluation metric (Higher is better)
> >
> > **Table 1: Analysis on the learned scaling with varying $\epsilon$**
> > | Initial $\epsilon$ | Learnable/Fixed | Final $\epsilon$ | Learned scale | Kendall's $\tau$ |
> > |------------|-----------------|--------------------|--------------------|--------------------|
> > | 0 (Baseline)      | -          | -             | -    | 0.922            |
> > | 0.001      | Fixed           | 0.001              | $1.000\pm0.001$    | 0.922            |
> > | 0.01       | Fixed           | 0.01              | $0.999\pm0.007$    | 0.923            |
> > | 0.1        | Fixed           | 0.1                | $1.003\pm0.079$    | 0.923            |
> > | 0.2        | Fixed           | 0.2                | $0.933\pm0.135$    | 0.923            |
> > | 0.4        | Fixed           | 0.4                | $0.972\pm0.378$   | 0.924            |
> > | 0.001      | Learnable       | 0.966              | $0.796\pm0.668$     | 0.925            |
> > | 0.01       | Learnable       | 0.970              | $0.941\pm0.710$     | 0.926            |
> > | 0.1        | Learnable       | 0.971              | $0.918\pm0.758$     | 0.926            |
> > | 0.2        | Learnable       | 0.946              | $0.743\pm0.626$   | 0.925            |
> > | 0.4        | Learnable       | 0.988              | $0.762\pm0.624$    | 0.925            |
> >
> >
> > When $\epsilon$ is fixed at a small value, the learned scaling slightly deviates from the identity, resulting in marginal performance gains. However, when $\epsilon$ is learnable, the learned scaling can deviate further from the identity, leading to a broader scaling range that enhances performance. Additionally, when $\epsilon$ is learnable, the final $\epsilon$ values tend to converge, regardless of their initial values. For larger values of $\epsilon$, instability may arise due to significant deviations in the early steps. Consequently, in our study, we choose a learnable $\epsilon$ with an initial value of 0.01 to ensure training stability.

---

> > > ### Author Response · Authors · 2025-11-19
> > >
> > > **Answer W4 (continue)**
> > >
> > > *Ablation on the $\epsilon$ in Transformers case*
> > >
> > > We present an ablation study on the sensitivity of $\epsilon$ by conducting the "Predict Transformer Generalization" experiment on the MNIST-Transformer dataset. Specifically, we analyze two cases: when $\epsilon$ is learnable versus fixed, and report the performance alongside statistics of learned noises ($\operatorname{sin}(\gamma(\theta)) \cdot \epsilon$) for the matrices $M$ and $N$, which represent the two $GL$ group actions applied to the weights of the MHA layer. For each noise, we compute the mean and standard deviation of the diagonal and off-diagonal elements. The results are summarized in Table 2 (Table 14 of Appendix B.8 in our revised version). The table reports the following values:
> > >
> > > - Initial $\epsilon$: The initial value of $\epsilon$ for both $M$ and $N$.
> > > - Learnable/Fixed: Whether $\epsilon$ is changed during training.
> > > - Final $\epsilon_M$, $\epsilon_N$: The final value of $\epsilon$ after training.
> > > - Diagonal $M$, $N$: Mean and standard deviation of the learned noises $\operatorname{sin}(\gamma(\theta)) \cdot \epsilon$ for $M$ and $N$, computed for diagonal elements.
> > > - Off-diagonal $M$, $N$: Mean and standard deviation of the learned noises $\operatorname{sin}(\gamma(\theta)) \cdot \epsilon$ for $M$ and $N$, computed for off-diagonal elements.
> > > - Kendall's $\tau$: Evaluation metric (higher is better).
> > >
> > > **Table 2: Sensitivity of $\epsilon$**
> > >
> > > |Initial $\epsilon$|Learnable/Fixed|Final $\epsilon_M$|Final $\epsilon_N$|Diagonal $M$|Off-diagonal $M$|Diagonal $N$|Off-diagonal $N$|Kendall's $\tau$|
> > > |-|-|-|-|-|-|-|-|-|
> > > |0 (Baseline)|-|-|-|-|-|-|-|0.905|
> > > |0.001|Fixed|0.001|0.001|$0.000\pm0.000$|$0.062\pm0.242$|$0.000\pm0.000$|$0.063\pm0.242$|0.909|
> > > |0.01|Fixed|0.01|0.01|$0.001\pm0.002$|$0.062\pm0.242$|$-0.001\pm0.005$|$0.062\pm0.242$|0.909|
> > > |0.1|Fixed|0.1|0.1|$0.005\pm0.020$|$0.062\pm0.245$|$-0.038\pm0.065$|$0.061\pm0.244$|0.907|
> > > |0.2|Fixed|0.2|0.2|$0.010\pm0.040$|$0.061\pm0.249$|$-0.002\pm0.144$|$0.059\pm0.285$|0.910|
> > > |0.4|Fixed|0.4|0.4|$0.021\pm0.080$|$0.060\pm0.265$|$-0.055\pm0.221$|$0.073\pm0.323$|0.909|
> > > |0.001|Learnable|0.001|-0.187|$0.000\pm0.000$|$0.063\pm0.242$|$-0.068\pm0.117$|$0.059\pm0.260$|0.908|
> > > |0.01|Learnable|0.010|0.182|$0.001\pm0.002$|$0.062\pm0.241$|$-0.056\pm0.084$|$0.061\pm0.244$|0.911|
> > > |0.1|Learnable|0.099|0.234|$0.005\pm0.020$|$0.062\pm0.245$|$-0.085\pm0.114$|$0.060\pm0.250$|0.910|
> > > |0.2|Learnable|0.200|0.290|$0.010\pm0.040$|$0.061\pm0.249$|$-0.011\pm0.218$|$0.062\pm0.316$|0.909|
> > > |0.4|Learnable|0.400|0.298|$0.021\pm0.079$|$0.060\pm0.265$|$-0.011\pm0.179$|$0.065\pm0.298$|0.909|
> > >
> > > When the $GL$ group is learned through our quasi-equivariant layers, the off-diagonal elements provide additional flexibility, leading to a noticeable improvement in performance even with a fixed $\epsilon$. With a learnable $\epsilon$, the range of values for $N$ expands and becomes more stable, resulting in better predictive performance. Overall, this study suggests that increasing the relaxation through $\epsilon$ can enhance performance. For our experiments, we select a learnable $\epsilon$ with an initial value of 0.01 to ensure training stability.

---

> > > > ### Author Response · Authors · 2025-11-19
> > > >
> > > > **Q1. The experiments are only focusing on invariant tasks. Why don't the authors perform experiments on equivariant tasks (e.g. INR editing)?**
> > > >
> > > > **Answer Q1.** Following your suggestion, we have run the INRs editing task following the settings in [1] and [2], and presented the results in Table 3 (Table 15 of Appendix B.9 in our revised version).
> > > >
> > > > **Table 3: Weight space style editing (INRs editing)**
> > > > |Model|Contrast (CIFAR-10)|Dilate (MNIST)|
> > > > |-|-|-|
> > > > |MLP|$0.031\pm0.001$|$0.306\pm0.001$|
> > > > |HNP[1]|$0.021\pm0.001$|$0.071\pm0.001$|
> > > > |NP[1]|$\underline{0.020\pm0.002}$|$\underline{0.068\pm0.002}$|
> > > > |Monomial-NFN[2]|$\underline{0.020\pm0.001}$|$0.069\pm0.002$|
> > > > |Monomial-NFN Quasi (1% additional parameters)|$\mathbf{0.019\pm0.001}$|$\mathbf{0.066\pm0.001}$|
> > > >
> > > > In this equivariant task, Monomial-NFN[2] initially falls behind NP[1] model. The result for Monomial-NFN is already the tuned version, and increasing the parameters cannot further improve performance. With our introduced Quasi layer that contains only 1% additional parameters, we can boost the performance of Monomial-NFN, making it surpass the performance of NP baseline.
> > > >
> > > > **Q2. The performance reported for baselines is often different from the original works (e.g. the works of Zhou et al., Kofinas et al.). In the case of INR classification, specifically, the performance gap is quite large. Why is that?**
> > > >
> > > > **Answer Q2.** For the INRs classification task, we follow the same settings as in Monomial-NFN[2], which utilized the dataset from [1]. The key difference between [1] and [2] is that [1] augments the data tenfold, resulting in 450,000 training instances, while [2] does not use augmentation. By not augmenting the dataset, they enable a fairer comparison, highlighting architectures that can inherently handle transformations in weight space without relying on additional data.
> > > >
> > > >
> > > > *References*
> > > >
> > > > [1] Zhou et al, Permutation Equivariant Neural Functionals. NeurIPS 2023.
> > > >
> > > > [2] Tran et al, Monomial Matrix Group Equivariant Neural Functional Networks. NeurIPS 2024.
> > > >
> > > > [3] Tran et al, Equivariant Neural Functional Networks for Transformers. ICLR 2025.
> > > >
> > > > [4] Unterthiner et al, Predicting Neural Network Accuracy from Weights. arxiv 2020.
> > > >
> > > > ---
> > > >
> > > > We thank the Reviewer for the constructive feedback and thoughtful suggestions. If our responses adequately address the concerns, we kindly hope that the evaluation may be adjusted to reflect this. We remain open to further discussion during the next stage of discussion.

---

### Official Review · Reviewer_7DXF · 2025-11-01

**Soundness:** 3
**Presentation:** 3
**Contribution:** 3
**Rating:** 6
**Confidence:** 3

**Summary:**

The paper concerns the efficient use of equivariance in the design of metanetworks, which are architectures intended to process the weights of another, predefined model. As different parameter values may identify the same functional mapping (the input-output relation of the model), it is a crucial advantage to identify and process functionally-equivalent sets of parameters.
This work in particular argues that exact equivariance is a too strict assumption to design effective metanetworks, while the weaker notion of quasi-equivariance is sufficient. It then introduces architectures that are able to implement this quasi-equivariance for feed-forward NNs and MHA, and demonstrates convincing results on a number of tasks.

**Strengths:**

- The topic is of current interest, and the paper does a convincing job in guiding the reader through computational and theoretical motivations for the work.
- The discussion around quasi-equivariance is interesting and convincing, and the notion is a novel contribution that has potentially wide application in the design of meta-networks.
- The discussion of the limitations (current limited applicability beyond linear architecture) is convincing.

**Weaknesses:**

**Definition and use of notion of quasi-maximality:**
- Definition 2.2: The sentence "Under generic parameters" is unclear. The entire informal definition seems to serve little purpose, as the formal one is given 10 lines after. I suggest Definition 2.2 to be simply stated in the text, without a "definition" environment.
Moreover, 1) the reason to consider epsilon to be a real algebraic variety is unclear here (instead of another set with other structure), and should be at least motivated if not clearly defined; 2) The notion of maximality seems to depend on $\varepsilon$, but it would be more clear if first it is stated without it (e.g., defining maximality and then $\varepsilon$-maximality).
- Section 2.2: "If G is a maximal symmetry group of the underlying model, such equivariance is sufficient to guarantee that F operates solely on the functional content of θ.". This again is not explicitly demostrated or discussed, and it is especially unclear why considering an algebraic variety is a sensible choice. This is partially motivated by the important example of MHA, which however comes only in Remark 4.4, several pages later.

**Well posedness of the notion of quasi-equivariance:**
- It is mentioned that "Further discussion and insights on this perspective are provided in Appendix A.1".
Since the conclusions of A.1 are quite involved, a short summary of these insights should be given in the main test. Positive and negative examples would be particularly helpful.
- Explanations would be appreciated also considering that this may guide practitioners in choosing or not this solution depending on the group of interest. Related to this, G is f-dependent. How does this relate to the results of Section A.1?
- The entire Section A.1 relies on heavy terminology and notions that are not tied to the rest of the paper and are not defined elsewhere. There should be either a concise introduction of the main tools, or at least references to the literature.

**Design of the quasi-equivariant map:**

The construction of the map $\alpha$, especially for MLPs (and specifically the part mapping to the diagonal matrices) is quite involved, and it is completely unclear why such an elaborate construction is required. Appendix B.1 does not help in understanding this. There should be some effort put into explaining why this construction is required in this crucial step.

**Questions:**

Apart from the points discussed above, there are the following minor points:
- Section 2.1: $f(\cdot; \theta)$ should be defined with a domain $D$, so that (1) has a precise meaning. Moreover, is $f$ at least continuous?

- Section 2.2: "...to produce incompatible outcomes". What does incompatible mean? So fa we are requiring $F(\theta)=F(\bar\theta)$ if the two parameters are in the same orbit (depend only on the underlying function represented by $\theta$).

- "Moreover, $G$-quasi-equivariance ensures functionality preservation": A one-line equation explaining this key point would be helpful.

- Section 4.1: The notion of continuity of a group-valued function should not be assumed to be common knowledge. Moreover, in Remark 4.1 the set $\Theta$ has connected components. But so far we always had $\Theta=R^{d}$. Is this no more the case? Moreover, for $\Theta=R^d$ this seems to imply that $\alpha$ is a constant function (see below for the matrix permutation group).

- Theorem 4.3: How reasonable are conditions 1/2 in practical, learned MHAs?

---

> ### Author Response · Authors · 2025-11-19
>
> We thank the Reviewer for the response and address the concerns below. The Reviewer may also refer to our General Response for a summary of the revisions.
>
> ---
>
> **W1. Definition and use of notion of quasi-maximality:**
>
> **- Definition 2.2: The sentence "Under generic parameters" is unclear. The entire informal definition seems to serve little purpose, as the formal one is given 10 lines after. I suggest Definition 2.2 to be simply stated in the text, without a "definition" environment. Moreover, 1) the reason to consider epsilon to be a real algebraic variety is unclear here (instead of another set with other structure), and should be at least motivated if not clearly defined; 2) The notion of maximality seems to depend on $\varepsilon$ , but it would be more clear if first it is stated without it (e.g., defining maximality and then $\varepsilon$-maximality).**
>
> **- Section 2.2: "If G is a maximal symmetry group of the underlying model, such equivariance is sufficient to guarantee that F operates solely on the functional content of θ.". This again is not explicitly demostrated or discussed, and it is especially unclear why considering an algebraic variety is a sensible choice. This is partially motivated by the important example of MHA, which however comes only in Remark 4.4, several pages later.**
>
> **Answer W1.** Our initial introduction of maximality was intentionally informal, aiming to convey the intuition arising from the motivating example (lines 134–145). This example illustrates the phenomenon that later appears formally in the characterization of architectural symmetries. However, we agree that placing an informal “definition” environment before the formal one may have caused confusion. In the revision, the informal explanation is now woven directly into the text, and the formal definition is kept as the only definition.
>
> *On the role of the proper real algebraic variety $\varepsilon$.* The set $\varepsilon$, defined as a proper real algebraic variety (i.e., the zero set of finitely many nonzero real polynomials), is not chosen arbitrarily. It aligns with the structure of pathological parameter configurations commonly excluded in prior works on symmetry characterization. For example:
>
> - an MLP weight matrix becomes rank-deficient,
> - two attention heads in an MHA block become exactly identical, or
> - certain blocks collapse to the zero operator.
>
> These pathological configurations constitute parameter sets of Lebesgue measure zero, and in all such cases the exceptional set can be expressed precisely as a proper real algebraic variety. This matches the assumptions used in previous symmetry analyses and ensures that the definition of maximality excludes only measure-zero degeneracies.
>
> *Why maximality is defined relative to $\varepsilon$.* The dependence on $\varepsilon$ is necessary for giving a formal and globally valid definition of maximality. Without excluding these degenerate configurations, the “maximal symmetry group” would be ill-defined: the symmetry group of the architecture could artificially enlarge at these exceptional points. This is highlight by two observations in Section 2 (line 134-150).

---

> > ### Author Response · Authors · 2025-11-19
> >
> > **W2. Well posedness of the notion of quasi-equivariance:**
> >
> > **- It is mentioned that "Further discussion and insights on this perspective are provided in Appendix A.1". Since the conclusions of A.1 are quite involved, a short summary of these insights should be given in the main test. Positive and negative examples would be particularly helpful.**
> >
> > **- Explanations would be appreciated also considering that this may guide practitioners in choosing or not this solution depending on the group of interest. Related to this, G is f-dependent. How does this relate to the results of Section A.1?**
> >
> > **- The entire Section A.1 relies on heavy terminology and notions that are not tied to the rest of the paper and are not defined elsewhere. There should be either a concise introduction of the main tools, or at least references to the literature.**
> >
> > **Answer W2.** We thank the Reviewer for this comment. The analysis addresses a potential confusion surrounding the relation $F(g\theta) = g' F(\theta) \text{ for some } g' \in G$, where $g'$ may depend on both $g$ and $\theta$.
> >
> > Because of this relation, a natural attempt to construct a $G$-quasi-equivariant map $F$ is to first choose an arbitrary group-valued function $\alpha : G \times \Theta \to G$, and then solve for $F : \Theta \to \Theta$ such that $F(g\theta) = \alpha(g,\theta)\, F(\theta)$.
> >
> > Such a map $F$ does not exist for a general choice of $\alpha$. The analysis in Appendix A.1 provides the conditions on $\alpha$ under which at least one corresponding $F$ can exist.
> >
> > However, even with this theoretical framework, we found that constructing $F$ through such an implicit constraint is not practical for metanetwork implementation. Instead, we adopt a simpler and more effective parameterization, $F = \alpha \beta$, where $\alpha : \Theta \to G$ and $\beta : \Theta \to \Theta$ is $G$-equivariant. Although this parameterization does not represent all possible quasi-equivariant metanetworks, it is efficient in practice and already leads to clear performance improvements.
> >
> > We thank the Reviewer once again and have revised the manuscript accordingly: the above discussion has been added to Sections 3 and 4, and a clarifying remark has been inserted in Appendix A.1. We kindly refer the Reviewer to these updated sections and would greatly appreciate any further suggestions to improve the manuscript.
> >
> > **W3. Design of the quasi-equivariant map: The construction of the map $\alpha$, especially for MLPs (and specifically the part mapping to the diagonal matrices) is quite involved, and it is completely unclear why such an elaborate construction is required. Appendix B.1 does not help in understanding this. There should be some effort put into explaining why this construction is required in this crucial step.**
> >
> > **Answer W3.** We thank the Reviewer for this comment. We provide a more detailed explanation below.
> >
> > In the MLP/CNN case, the map $\alpha$ consists of two parts: a constant map for the group $\mathcal{P}\_n$ and a map for the group $\mathbb{R}\_{>0}^n$. To learn the latter map, we construct a network that can translate from the weight space to a diagonal matrix in $\mathbb{R}\_{>0}^n$, which can also be represented as a vector in $\mathbb{R}^{n}$ with all positive entries. We compute statistical features of the weights, with shape $D$, to serve as input to the network. Therefore, the introduced network is basically a MLP $\{W{\text{scale}}, b_{\text{scale}}\}$ that maps from $\mathbb{R}^D \to \mathbb{R}^n$.
> >
> > The formula $\operatorname{sin}(W_{\text{scale}} x + b_{\text{scale}}) \cdot \epsilon + 1_n$ naturally relaxes the strict equivariance typically imposed in metanetworks. By introducing a sine function, we allow the transformation to gently oscillate around the identity, creating a smooth, controlled variation. This approach is inspired by Fourier analysis, where sine waves naturally introduce periodic fluctuations, offering flexibility without disrupting the overall structure. The small learnable parameter $\epsilon$ simply scales these oscillations, determining how much relaxation to apply. This design provides a natural way to balance stability and flexibility, enabling the model to remain expressive while avoiding the constraints of strict equivariance.
> >
> > We have incoporated this explanation to the end of Appendix B.1 (line 1144-1156) in our revised manuscript.

---

> ### Author Response · Authors · 2025-11-19
>
> **Q1. Section 2.1: $f(\cdot;\theta)$ should be defined with a domain , so that (1) has a precise meaning. Moreover, is $f$ at least continuous?**
>
> **Answer Q1.** We thank the Reviewer for the suggestion. The function $f$ is simply a parameterized function with parameter $\theta$. The notation $f(\,\cdot\,;\theta)$ is standard, where “\(\cdot\)” denotes the input argument of $f$. The domain of $f$ can be arbitrary and does not affect any of the definitions that follow.
>
> $f$ does not need to be continuous; it may be any function. For clarity, we provide several examples:
>
> - $f : \mathbb{R} \to \mathbb{R}$ with $\theta = (a,b,c) \in \mathbb{R}^3$, where  $f(x; a,b,c) = ax^3 + bx + c$.
> - $f : \mathbb{R} \to \mathbb{R}$ with $\theta = (a,b) \in \mathbb{R}^2$, where  $f(x; a,b) = \lfloor ax + b \rfloor$,
>     which is not continuous.
> - $f : \mathbb{Q} \to \mathbb{R}$ with $\theta = (a,b) \in \mathbb{R}^2$, where  $f(x; a,b) = \operatorname{sign}(ax^2 + b)$ which has $\mathbb{Q}$ as its domain.
>
> We hope that this explanation and the examples address the Reviewer's concern.
>
> **Q2. Section 2.2: "...to produce incompatible outcomes". What does incompatible mean? So far we are requiring $F(\theta)=F(\bar\theta)$ if the two parameters are in the same orbit (depend only on the underlying function represented by $\theta$).**
>
> **Answer Q2.** We address the Reviewer's concern as follows. Recall the operator that assigns each parameter $\theta$ to its corresponding function
>
> $\rho \colon \Theta \to \{\text{the set of all functions that the model } f \text{ represents}\},
> \qquad
> \theta \mapsto f(\cdot;\theta),$
>
> where $\rho$ is exactly the same as the map $[\cdot]$ in Eq.(1).
>
> Let $F$ be the metanetwork. The model that we train is $F$ itself, but the model we ultimately need is $\rho \circ F$, since $\rho$ cannot be included in training due to its discrete nature. Intuitively, $F$ is a parameterized, vector-to-vector map that computers can process, whereas $\rho$ assigns a parameter vector to the corresponding function, which is not differentiable and therefore cannot be optimized directly.
>
> The content the Reviewer refers to is in Section 2.2, where we discuss the necessity of equivariance in metanetworks. Let $\theta$ and $\bar{\theta}$ lie in the same orbit, i.e., $[\theta] = [\bar{\theta}]$. Our requirement is *not* that $F(\theta) = F(\bar{\theta})$, but rather that $[F(\theta)] = [F(\bar{\theta})]$,  meaning that the functions represented by $F(\theta)$ and $F(\bar{\theta})$ are identical, as they correspond to parameters in the same equivalence class.
>
> In other words, if $G$ is a maximal symmetry group, then $[\theta] = [\bar{\theta}]$ implies $\bar{\theta} = g\theta$ for some $g \in G$. If $F$ is equivariant, then $F(g\theta) = g F(\theta)$, which directly yields $[F(\theta)] = [F(\bar{\theta})]$. This illustrates why equivariance arises as a necessary property in the design of metanetworks.
>
>
> **Q3. "Moreover, $G$-quasi-equivariance ensures functionality preservation": A one-line equation explaining this key point would be helpful.**
>
> **Answer Q3.** The sentence "Moreover, $G$-quasi-equivariance ensures functionality preservation" means the following: for any $\theta, \bar{\theta} \in \Theta$, if $F:\Theta \to \Theta$ is $G$-quasi-equivariant, then whenever $[\theta] = [\bar{\theta}]$, we must also have $[F(\theta)] = [F(\bar{\theta})]$.
>
> Since $G$ is a maximal symmetry group, the condition $[\theta] = [\bar{\theta}]$ implies that $\bar{\theta} = g\theta$ for some $g \in G$. Because $F$ is $G$-quasi-equivariant, we have $F(g\theta) = g' F(\theta)$ for some $g' \in G$. This immediately yields $[F(\theta)] = [F(\bar{\theta})]$, which is the desired functionality preservation.

---

> ### Author Response · Authors · 2025-11-19
>
> **Q4. Section 4.1: The notion of continuity of a group-valued function should not be assumed to be common knowledge. Moreover, in Remark 4.1 the set $\Theta$ has connected components. But so far we always had $\Theta=R^d$. Is this no more the case? Moreover, for $\Theta=R^d$ this seems to imply that \alpha is a constant function (see below for the matrix permutation group).**
>
> **Answer Q4.** We thank the Reviewer for the suggestion and have added the necessary details in our revision. This point is clarified in Section 4.1, where we set up the framework for designing quasi-equivariant networks.
>
> Recall a standard topological fact: if $f : X \to Y$ is continuous, $X$ is connected (i.e., has a single connected component), and $Y$ is equipped with the discrete topology, then $f$ must be constant.
>
> In our setting, $X = \Theta \subseteq \mathbb{R}^d$, which is always a connected space. If $Y$ is the permutation group, then $Y$ carries the discrete topology. Consequently, any continuous group-valued map $\alpha : \Theta \to Y$ must be constant. This means that the discrete part of the symmetry group (e.g., permutations) does not contribute a meaningful transformation for the purpose of defining quasi-equivariance.
>
> For this reason, our analysis focuses only on the continuous components of the maximal symmetry group: the positive scaling group (for feedforward and convolutional networks) and the general linear group (for multihead attention). These are the components for which nontrivial continuous $\alpha$ maps exist and thus are relevant for constructing $G$-quasi-equivariant metanetworks.
>
>
> **Q5. Theorem 4.3: How reasonable are conditions 1/2 in practical, learned MHAs?**
>
> **Answer Q5.** The conditions in Theorem 4.3 generally hold in practice, for the following reasons:
>
> - For a matrix of size $d_h \times d$ with $d_h < d$, after training, the matrix almost surely attains full row rank $d_h$, since the set of matrices with rank strictly smaller than $d_h$ has measure zero in $\mathbb{R}^{d_h \times d}$.
> - For any finite collection of real matrices, training almost surely results in them being pairwise distinct, because the set of parameter configurations where two matrices coincide is also of measure zero.
>
> Assumptions that exclude a measure-zero subset of the parameter space -- often formalized, as in our paper, by describing this subset as a proper real algebraic set} (the zero set of finitely many nonzero real polynomials)-- are standard in prior works analyzing symmetry in neural architectures, such as [1], [2], [3], [4].
>
>
> *References*
>
> [1] Robert Hecht-Nielsen. On the algebraic structure of feedforward network weight spaces
>
> [2] Charles Fefferman and Scott Markel. Recovering a feed-forward net from its output
>
> [3] Phuong Bui Thi Mai and Christoph Lampert. Functional vs. parametric equivalence of relu networks
>
> [4] Tran et al., On Linear Mode Connectivity of Mixture-of-Experts Architectures
>
> We hope this clarifies the Reviewer's concern.
>
> ---
>
> We thank the Reviewer for the constructive feedback and thoughtful suggestions. If our responses adequately address the concerns, we kindly hope that the evaluation may be adjusted to reflect this. We remain open to further discussion during the next stage of discussion.

---

> > ### Comment · Reviewer_7DXF · 2025-11-24
> > **Response to the authors**
> >
> > I thank the authors for their careful and detailed rebuttal. The clarifications provided, together with the updates made to the manuscript, address the main points I had raised. While I still see a few aspects of the presentation that could be further refined, these are relatively minor and do not substantially affect my overall assessment. The additional experiments and explanations also improved my confidence in the strength and completeness of the contribution.
> > In light of this, I am increasing my score from 6 to 8.
> > As a small note: in line 1149 in the revised PDF there is a "_" missing.

---

> > > ### Author Response · Authors · 2025-11-24
> > > **Thank you for your endorsement!**
> > >
> > > We thank the Reviewer for the response and appreciate the re-evaluation of our submission. We also thank the Reviewer for raising these points, which we have addressed with the necessary revisions. As the discussion phase is still ongoing, we remain open to further discussion should the Reviewer have any additional concerns.

---

> > > > ### Author Response · Authors · 2025-11-25
> > > >
> > > > Dear Reviewer 7DXF,
> > > >
> > > > Once again, we sincerely thank you for your valuable feedback and for your decision to revise the score of our submission from 6 to 8. However, we noticed that the score in the official review has not yet been updated. While we hope we have not misunderstood anything, we kindly ask whether you could update the score on the official review.
> > > >
> > > > Thank you very much.
> > > >
> > > > Best regards,
> > > > The Authors

---

### Official Review · Reviewer_ya8b · 2025-11-05

**Soundness:** 3
**Presentation:** 3
**Contribution:** 3
**Rating:** 6
**Confidence:** 3

**Summary:**

This work proposes a novel theoretical framework for relaxing the rigidity of strict equivariant meta-networks while still preserving functional identity. The authors formalize this relaxed notion of equivariance as quasi-equivariance and propose a general construction where a learned group action $\alpha(\theta)$ modulates an already equivariant backbone. After describing how they can implement such a network in the case of MLPs, CNNs, and Transformers, they apply it in generalization prediction tasks and the INR image classification task. Across all architectures, the proposed relaxation of the strict equivariance seems to provide small improvements in the overall performance without a significant increase in the number of parameters.

**Strengths:**

- The framework of quasi-equivariant networks and their connection to the maximal symmetry group and the preservation of the functional identity are presented with significant mathematical rigor. As such, it provides a solid theoretical foundation and a promising starting point for the community to further investigate the proposed framework and its properties.
- The proposed framework and the decomposition of quasi-equivariant networks into $F(\theta)=\alpha(\theta)\beta(\theta)$ is quite general, allowing the approach to be implemented for a variety of tasks and network architectures. The authors partially demonstrate this adaptability by applying it to the most commonly used architectures.
- The experimental evaluation shows that the proposed network provides consistent improvement over the baseline equivariant network without a significant increase in the computational complexity of the method.

**Weaknesses:**

- The authors claim that introducing the $\alpha(\theta)$ improves the expressivity of the networks, but they do not provide sufficient  evidence to support such a claim. Specifically, if the task or loss we are interested in is invariant to the group transformation introduced by $\alpha$, does the preservation of the functional identity imply that the proposed quasi-equivariant architecture will have the same expressivity as the original one?
- The experimental section lacks ablation studies comparing the proposed approach with other methods that also relax exact equivariance. Thus, it is difficult to conclude whether the improvements in performance come from the specific framework of the quasi-equivariant network or are more generally caused by the relaxation of the equivariant constraint.

**Questions:**

- If the loss used to train the quasi-equivariant networks is invariant to the transformation introduced by $\alpha(\theta)$, since both the original and transformed parameters map to the same function, does this imply that the loss gradient with respect to $\alpha(\theta)$ is zero? If so, does this phenomenon occur in the current experimental setup?
- How do other approximate or relaxed equivariant methods compare to the proposed quasi-equivariant networks?

---

> ### Author Response · Authors · 2025-11-19
>
> We thank the Reviewer for the response and address the concerns below. For clarity and coherence, we found it appropriate to merge certain related weaknesses and questions and provide unified answers. We first recall several relevant points from our work, and then address the Reviewer's concerns in detail. The Reviewer may also refer to our General Response for a summary of the revisions.
>
> ---
>
> Recall the operator that assigns each parameter $\theta$ to its corresponding function
>
> $\rho \colon \Theta \to \{\text{the set of all functions that the model } f \text{ represents}\},
> \qquad
> \theta \mapsto f(\cdot;\theta),$
>
> where $\rho$ is exactly the same as the map $[\cdot]$ in Eq.(1).
>
> Let $F$ be the metanetwork. The model that we train is $F$ itself, but the model we ultimately need is $\rho \circ F$, since $\rho$ cannot be included in training due to its discrete nature. Intuitively, $F$ is a parameterized, vector-to-vector map that computers can process, whereas $\rho$ assigns a parameter vector to the corresponding function, which is not differentiable and therefore cannot be optimized directly.
>
> ---
>
> **W1. The authors claim that introducing the $\alpha(\theta)$ improves the expressivity of the networks, but they do not provide sufficient evidence to support such a claim. Specifically, if the task or loss we are interested in is invariant to the group transformation introduced by $\alpha$, does the preservation of the functional identity imply that the proposed quasi-equivariant architecture will have the same expressivity as the original one?**
>
> **Answer W1.** In Equation (4), the appearance of the map $\alpha \colon \Theta \to G$ does increase the expressivity of the usual equivariant map $\beta$, since, in principle, the composition $F = \alpha \beta$ represents a larger function class than $\beta$ alone. The "same expressivity" mentioned by the Reviewer refers to the function class of $\rho \circ F$, because $\rho \circ F$ and $\rho \circ \beta$ are indeed identical.
>
> The model we train, however, is $F$, and this additional expressivity leads to better empirical performance. We would also like to share a related phenomenon that supports this observation: in the training of attention mechanisms, the parameters of the query and key networks typically tend to remain balanced. During training, if we artificially apply transformations to the parameters that induce imbalance -- even if the represented function remains unchanged -- the optimization dynamics worsen significantly. This illustrates why a more expressive map $F$ can be beneficial in practice, even though $\rho \circ F$ and $\rho \circ \beta$ correspond to the same underlying function.
>
> We hope this explanation addresses the Reviewer's concern.
>
> **Q1. If the loss used to train the quasi-equivariant networks is invariant to the transformation introduced by $\alpha(\theta)$, since both the original and transformed parameters map to the same function, does this imply that the loss gradient with respect to $\alpha(\theta)$ is zero? If so, does this phenomenon occur in the current experimental setup?**
>
> **Answer Q1.**  As explained in **W1**, the model being trained is $F$, not $\rho \circ F$. This distinction is essential. While $\rho \circ F$ is invariant under the transformation induced by $\alpha(\theta)$, the map $F = \alpha \beta$ itself is not. Thus, changing $\alpha$ alters the parameterization of $F$ in a nontrivial way, and the loss is evaluated on the output of $F$ rather than on the invariant function $\rho \circ F$. For this reason, the loss gradient with respect to $\alpha(\theta)$ is not zero.

---

> ### Author Response · Authors · 2025-11-19
>
> **W2. The experimental section lacks ablation studies comparing the proposed approach with other methods that also relax exact equivariance. Thus, it is difficult to conclude whether the improvements in performance come from the specific framework of the quasi-equivariant network or are more generally caused by the relaxation of the equivariant constraint.**
>
> **Q2. How do other approximate or relaxed equivariant methods compare to the proposed quasi-equivariant networks?**
>
> **Answer W2+Q2.** The reason we do not compare or implement other ideas on relaxed equivariance is as follows.
>
> In previous work, the metanetwork $F$ is designed to be equi/invariant. However, the actual equi/invariance property we desire lies in $\rho \circ F$, a notion we call *preserve functionality*. This is the main motivation for introducing *quasi-equivariance* (Section 2.2). Importantly, quasi-equivariance of $F$ is *necessary and sufficient* for $\rho \circ F$ to satisfy preserve functionality (lines 204-209). We also emphasize that quasi-equivariance is meaningful only in the context of metanetworks or other applications where the relevant map is of the form $\rho \circ F$.
>
> We now return to the two relaxed-equivariance notions mentioned in Remark 3.4.
>
> - $\varphi(gx) = g' \varphi(x)$ for some $g' \in G_x$.
> This variant is already subsumed by our definition. The additional requirement $g' \in G_x$ (instead of $g' \in G$) does not play any meaningful role and therefore does not introduce a distinct or useful relaxation in our setting.
> - $\varphi(gx) \approx g\varphi(x)$.
> At first sight, this approximation could be applied to metanetworks. However, the approximation concerns $F$ itself. After composing with $\rho$, the resulting map $\rho \circ F$ generally cannot achieve any reasonable form of "functional-preserving approximation", because this depends heavily on how the model $f$ is parameterized. For instance, in an MLP $f(\cdot;\theta)$, having $\theta \approx \theta'$ does not imply that $f(\cdot;\theta)$ approximates $f(\cdot;\theta')$ in any meaningful sense, unless one further analyzes the continuity or stability properties of the parametrization. A secondary issue is that approximate equivariance in the literature typically concerns simple groups such as $S_n$, $\mathrm{SO}(n)$, $\mathrm{O}(n)$, or $\mathrm{GL}(n)$, whereas the symmetry groups of neural architectures are often large products of groups. This makes such relaxed notions either inefficient or inapplicable in the metanetwork setting.
>
> For these reasons, we do not adopt or implement other forms of relaxed equivariance in our metanetwork framework. We hope this clarifies and adequately addresses the Reviewer's concern.
>
> ---
>
> We thank the Reviewer for the constructive feedback and thoughtful suggestions. If our responses adequately address the concerns, we kindly hope that the evaluation may be adjusted to reflect this. We remain open to further discussion during the next stage of discussion.

---

### Author Response · Authors · 2025-11-19
**General Response**

Dear AC and reviewers,

Thank you for your thoughtful reviews and valuable feedback, which have greatly helped us enhance the paper.

We sincerely appreciate the reviewers for their insightful comments and constructive suggestions. We are encouraged by the positive remarks that:

1. The topic is novel and of current interest (Reviewer 7DXF, 9pd6), offering a significant extension to the existing paradigm of strict equivariance (Reviewer J9ur).

2. The paper is well written and well-structured (Reviewer 7DXF, 9pd6, J9ur). The introduced framework is presented with strong mathematical rigor, providing a solid theoretical foundation for further research (Reviewer ya8b, 9pd6, J9ur).

3. The proposed method demonstrates adaptability to diverse architectures, bringing consistent improvements without a significant increase in computational complexity (Reviewer ya8b, 9pd6, J9ur).

Incorporating the Reviewers' comments and suggestions, as well as additional empirical studies that we believe are informative, we summarize below the main revisions made to the manuscript (all changes are highlighted in red).

1. Add further clarification in the definition of Maximal symmetry group (lines 155-159), Remark 3.2 (lines 216-222), Remark 4.1 (lines 272-279), extend the discussion for further applications (line 514-519).

2. Add further clarification in the extension to CNN (Remark 4.2, line 324-332) and the design of quasi-equivariant layer (line 1144-1156)

3. Conduct an equivariant experiment - Weight space style editing (Appendix B.9).

4. Conduct two ablation studies on the sensitivity of $\epsilon$ and learned scaling (Appendices B.7 and B.8).

5. Fix minor issues (citation style (line 66, 70), linear group introduction (line 354), table 3 formatting (line 486-504)) for improved clarity.

---

### Author Response · Authors · 2025-12-01
**Submission Briefing for the New AC: Thank you for Taking on the Oversight of Our Submission**

Dear new AC,

Thank you for stepping in at this challenging stage of the review process and taking over the handling of our submission. We are truly grateful for your time and support.

To make it easier for you to review the paper, we have prepared three short summaries, shared in the messages below:

**(Briefing 1) Summary of Our Key Contributions**

**(Briefing 2) Summary of Main Concerns of Reviewers and Our Replies**

**(Briefing 3) Summary of Individual Concerns and Each Reviewer’s Reply to Our Rebuttal**

We submitted our rebuttal on November 19 (AoE). Despite several reminders, only Reviewers 7DXF and J9ur participated in the discussion phase. We are grateful for their input, particularly their shared view that our rebuttal was **careful and detailed** and **resolved their main concerns**. They also highlighted the **strength of our experimental results**, which increased their confidence in the work; accordingly, Reviewer 7DXF **raised their score from 6 to 8** on November 24 (AoE).

We would kindly ask you to review our paper, rebuttal, and subsequent responses with additional results, bearing in mind that we have systematically addressed all points raised by the reviewers. We believe the remaining reviewers would likely have revised their assessments had the discussion continued.

We are confident that, with your careful consideration, our submission will receive a fair and accurate evaluation by the AC, SAC, and PC. Please feel free to contact us with any questions about the submission or rebuttal-we would be glad to provide further clarification or continue the discussion.

Best regards,

Authors

---

> ### Author Response · Authors · 2025-12-01
> **Briefing 1: Summary of Our Key Contributions.**
>
> Dear Area Chair,
>
> To facilitate your evaluation, we provide a concise summary of the main contributions and core theoretical and experimental results of our submission “Quasi-Equivariant Metanetworks.” This message is intended as a high-level roadmap to the paper.
>
> **Motivation and Novel Concept**
>
> 1. We identify a fundamental limitation of strict parameter-space equivariance in metanetworks: the parameterization of a neural network is not injective with respect to the function it represents, so different weight configurations may encode the same function.
>
> 2. To address this, we introduce quasi-equivariance, a relaxed symmetry notion that preserves functional identity under architectural transformations while avoiding the rigidity of strict equivariance.
>
> 3. The goal is to enable metanetworks to respect architectural symmetries at the level of functional equivalence classes, rather than being constrained by strict weight-space orbits.
>
> **Main Theoretical Contributions**
>
> 1. We formally define quasi-equivariance with respect to a maximal symmetry group of the architecture, and show that it provides a necessary and sufficient condition for functionality preservation, i.e., it guarantees that metanetworks act consistently on functional equivalence classes rather than individual parameterizations.
>
> 2. We prove that quasi-equivariance strictly generalizes classical equivariance and yields a strictly larger hypothesis class, while remaining compatible with the symmetry constraints required for principled metanetwork design (including closure under composition and integration with invariant heads).
>
> **Experimental Contributions**
>
> 1. We provide detailed implementations of the quasi-equivariant layer for feedforward, convolutional, and Transformer-based architectures, and integrate it into existing metanetworks (Monomial-NFN and Transformer-NFN).
>
> 2. Across three representative benchmarks-CNN generalization prediction from weights, INR classification, and Transformer generalization prediction-quasi-equivariant metanetworks consistently outperform strictly equivariant baselines while handling symmetry groups appropriately.
>
> 3. With only a minimal number of additional parameters, the proposed layer functions as an efficient yet effective plug-in module, delivering substantial performance gains with negligible model-size overhead.

---

> ### Author Response · Authors · 2025-12-01
> **Briefing 2: Main Concerns of Reviewers and Our Replies.**
>
> Dear Area Chair,
>
> In this message, we would like to summarize the key concerns raised by the Reviewers.
>
> **Concern 1: Clarification on specific topics**. The reviewers asked for some clarification in order to make the paper more understandable for the non-expert in group theory, or to provide more explanation on precise details that were maybe omitted in our first submission.
>
> **Our Rebuttal:**
> - **Detail on how $\alpha$ is improving the expressivity of the networks** (Reviewer 8ayb). We explained that $\alpha$ was improving the expressivity of the networks since the composition $F = \alpha \beta$ represents a larger function class than $\beta$ alone. Moreover we specified that the model being trained is $F$, not $\rho \circ F$. And this distinction is essential. While $\rho \circ F$ is invariant under the transformation induced by $\alpha(\theta)$, the map $F = \alpha \beta$ itself is not. Thus, changing $\alpha$ alters the parameterization of $F$ in a nontrivial way, and the loss is evaluated on the output of $F$ rather than on the invariant function $\rho \circ F$. For this reason, the loss gradient with respect to $\alpha(\theta)$ is not zero.
>
> - **Extra precision about the definition of functionality preservation** (Reviewer 8ayb). We added explanation about functionality preservation: for any $\theta, \bar{\theta} \in \Theta$, if $F:\Theta \to \Theta$ is $G$-quasi-equivariant, then whenever $[\theta] = [\bar{\theta}]$, we must also have $[F(\theta)] = [F(\bar{\theta})]$.
>
> **Concern 2: Regarding the experiments**: The reviewers questioned the intuition behind the design of our method and requested additional experiments.
>
> **Our Rebuttal:**
>
> - **Further details on the design of the map $\alpha$** (Reviewer 9pd6 and 7DXF). We explained that $\alpha$ has two components: a constant map for $\mathcal{P}^n$ and a learnable map for $\mathbb{R}{>0}^n$. The latter is implemented by a network mapping from weight space to a diagonal matrix in $\mathbb{R}{>0}^n$, parameterized as $\sin(W_{\text{scale}} x + b_{\text{scale}})\cdot \epsilon + \mathbf{1}_n$. The sine term allows smooth oscillations around the identity, while the small learnable $\epsilon$ controls their magnitude. This yields a controlled relaxation of strict equivariance that balances stability with expressivity.
>
> - **Additional Ablation studies** (Reviewer 9pd6 and ya8b): Reviewers asked for more analysis of the learned group elements, especially the sensitivity of $\epsilon$ and the scaling factors. We added two ablation studies on CNNs and Transformers, varying $\epsilon$ in both fixed and learnable settings. When $\epsilon$ is fixed and small, the learned scaling stays close to the identity, yielding only modest gains. When $\epsilon$ is learnable, the scaling deviates more from the identity, enlarging the effective scaling range and improving performance. In this case, $\epsilon$ also converges to similar values regardless of its initialization.
>
> - **Compare with other relaxed or approximated equivariant models** (Reviewer ya8b, J9ur): We clarified why we do not compare against other relaxed equivariance notions. Our proposed quasi-equivariance is necessary and sufficient to ensure $\rho \circ F$ preserves functionality. Alternative definitions are either already subsumed by our framework or rely on approximate equivariance ($\varphi(gx) \approx g\varphi(x)$), which is unsuitable here because approximating parameters does not guarantee approximating the resulting model's function. Furthermore, existing approximate methods typically target simple groups (e.g., $S_n$, $\mathrm{SO}(n)$) and are inefficient for the large, complex product groups inherent to neural architectures.
>
>
> Please refer to our point-by-point rebuttal for each reviewer for the full explanations and experimental results. All additional clarifications and results have been incorporated into the revised manuscript and are listed in our previous General Response.

---

> ### Author Response · Authors · 2025-12-01
> **Briefing 3: Summary of Additional Theoretical and Empirical Results during the Rebuttal & Discussion Period.**
>
> During the rebuttal, we engaged closely with all reviewers, providing clarifications, new theoretical insights, and additional experiments that strengthened our submission. Below we summarize, reviewer by reviewer, how we addressed their specific concerns (beyond the common ones) and how the reviewers reacted, including cases where they increased their confidence or raised their scores.
>
> ---
>
> **Reviewer ya8b**
>
> **Concern 1. Additional elaboration on how introducing the quasi-equivariant relaxation increases expressivity, and how we obtain non-zero gradients.**
>
> **Concern 2. Other relaxed equivariance methods.**
>
> **Our Rebuttal**: Both are common concerns and are addressed in detail in our rebuttal.
>
> ---
>
> **Reviewer 7DXF**
>
> **Concern 1. Clarity and well-posedness of maximality, quasi-equivariance, and functionality preservation**
>
> **Our Rebuttal to Concern 1**: We kept only the formal definition of maximality, motivated $\varepsilon$ as a proper real algebraic variety capturing standard measure-zero degeneracies, and summarized the key insights of Appendix A.1 (with examples and guidance) in the main text. We also made functionality preservation explicit by stating that for a $G$-quasi-equivariant $F$, $[\theta] = [\bar{\theta}]$ implies $[F(\theta)] = [F(\bar{\theta})]$, and adopted the practical parameterization $F = \alpha \circ \beta$ with $G$-equivariant $\beta$.
>
> **Concern 2. Practicality of topological and algebraic assumptions (continuity, permutations, Theorem 4.3)**
>
> **Our Rebuttal to Concern 2**: We clarified that any continuous map $\alpha : \Theta \to$ (permutation group) must be constant when $\Theta \subseteq \mathbb{R}^d$ is connected, which justifies focusing quasi-equivariance on the continuous symmetry components (scalings and $\mathrm{GL}$-type groups). We further explained that the rank and distinctness conditions in Theorem 4.3 and the exclusion of a proper real algebraic variety hold almost surely after training and are standard in symmetry analyses, showing that our assumptions are realistic for practical MHAs.
>
> **Concern 3. Additional explanation on the construction of quasi-equivariance**
>
> **Our Rebuttal to Concern 3**: We added a detailed explanation of the construction of $F$ and $\alpha$ in Appendix B.1.
>
> **$\Rightarrow$ Reviewer reply**: The reviewer thanked us for a **careful and detailed** rebuttal and stated that the clarifications and changes in the revised manuscript **addressed their main concerns**. They noted that the remaining issues are minor presentation points that do not affect their overall assessment, and that the additional experiments and explanations **increased their confidence** in our contribution, so they **raised their score from 6 to 8**.
>
> ---
>
> **Reviewer 9pd6**
>
> **Concern 1. Lack of clarity, namely the design of the quasi-equivariant layer**
>
> **Our Rebuttal to Concern 1**: We added a detailed explanation of the layer design and construction in Appendix B.1.
>
> **Concern 2. Experimental benchmarks and datasets**
>
> **Our Rebuttal to Concern 2**: We followed the experimental setups from recent metanetwork research, utilizing popular datasets: Small CNN Zoo (2020), SIRENs dataset (2023), and Small Transformer Zoo (2025).
>
> **Concern 3. More ablation studies and equivariant tasks**
>
> **Our Rebuttal to Concern 3**: We conducted two ablation studies in Appendices B.7 and B.8. For the equivariant task, we followed previous work and ran INR-editing experiments on CIFAR-10 Contrast and MNIST Dilate (Appendix B.9), showing that our layer enables metanetworks from prior work to surpass their baselines.
>
> ---
>
> **Reviewer J9ur**
>
> **Concern 1. Need for elaboration when applying to more complex architectures like graph-based metanetworks and future possible applications**
>
> **Our Rebuttal to Concern 1**: We explained that extending quasi-equivariance to graph-based metanetworks is nontrivial because current graph metanetworks mostly support permutation symmetries and lack practical mechanisms for broader symmetries, so we leave this as future work. We also extended the conclusion to include a discussion of possible applications.
>
> **Concern 2. More analysis on $\epsilon$ and learned scaling factors**
>
> **Our Rebuttal to Concern 2**: We conducted two ablation studies in Appendices B.7 and B.8 analyzing $\epsilon$ and the learned scaling factors.
>
> **$\Rightarrow$ Reviewer Reply**: The reviewer **appreciated our careful revisions and responses to each reviewer, including detailed experimental results**. They suggested adding these explanations to the paper and asked whether non-metanetwork relaxed equivariance methods can be applied in our setting.
>
> **Answer to Reviewer’s further question**: This was fully answered in our reply to Reviewer ya8b.

---

### Meta-Review · Area_Chair_HjrX · 2025-12-28

**Summary:**

The paper introduces quasi-equivariance, a principled relaxation of strict equivariance for metanetworks—neural architectures that operate on the weights of other networks. The key insight is that the parameter-to-function mapping is non-injective (different weights can represent identical functions), so strict equivariance is overly rigid. Quasi-equivariance requires that for any group transformation g applied to input θ, the output is transformed by some (possibly different) group element α(θ,g), preserving functional identity while allowing greater expressivity. The authors provide theoretical foundations connecting this to maximal symmetry groups, propose a practical construction via learned group actions composed with equivariant maps, and demonstrate consistent improvements across feedforward, convolutional, and transformer architectures with minimal parameter overhead. All reviewers acknowledged the novelty of the quasi-equivariance concept and the solid theoretical foundation, with positive remarks on writing quality and experimental breadth.

**Reviewer Concerns:**

The authors provided a thorough rebuttal addressing all major concerns. Reviewers 7DXF and J9ur engaged constructively and confirmed their concerns were resolved—7DXF explicitly raised their score from 6 to 8. Key additions include detailed explanations of the layer design, two new ablation studies on ε sensitivity and learned scaling factors (Appendices B.7, B.8), and an equivariant INR-editing experiment.

**Reviewer Scores:**

Reviewer 7DXF updated their score to 8 as stated. Reviewer J9ur maintained their positive score of 8. Given the detailed responses to ya8b's concerns about expressivity and the comparison with relaxed equivariance methods (which the authors convincingly argued are either subsumed or inapplicable), ya8b would likely have maintained or slightly increased their score. Similarly, 9pd6's concerns about ablations and clarity were directly addressed with new experiments and explanations; a modest score increase would be reasonable had they engaged.

---

### Decision · Program_Chairs · 2026-01-26

Accept (Poster)